



# Local to regional methane emissions from the Upper Silesia Coal Basin (USCB) quantified using UAV-based atmospheric measurements

Truls Andersen[1], Marcel de Vries[1], Jaroslaw Necki[4], Justyna Swolkien[5], Malika Menoud[6], Thomas Röckmann[6], Anke Roiger[7], Andreas Fix[7], Wouter Peters[1,3], and Huilin Chen[1,2*]

[1]Centre for Isotope Research, Energy and Sustainability Institute Groningen (ESRIG), University of Groningen, Groningen, Netherlands

[2]Joint International Research Laboratory of Atmospheric and Earth System Sciences, School of Atmospheric Sciences, Nanjing University, Nanjing, China

[3]Meteorology and Air Quality, Wageningen University and Research Center, Wageningen, Netherlands

[4]Faculty of Physics and Applied Computer Science, AGH University of Science and Technology, Krakow, Poland

[5]Faculty Civil Engineering and Resource Management, AGH University of Science and Technology, Krakow, Poland

[6]Institute for Marine and Atmospheric Research Utrecht (IMAU), Utrecht University, Utrecht, Netherlands

[7]Deutsches Zentrum für Luft- und Raumfahrt e.V. (DLR), Institut für Physik der Atmosphäre, Oberpfaffenhofen, Germany

*Correspondence to*: Huilin Chen (huilin.chen@rug.nl)

**Abstract.** Coal mining accounts for ~ 12 % of the total anthropogenic methane emissions worldwide. The Upper Silesian Coal Basin, Poland, where large quantities of $CH_4$ are emitted to the atmosphere via ventilation shafts of underground hard coal (anthracite) mines, is one of the hot spots of methane emissions in Europe. However, coalbed $CH_4$ emissions into the atmosphere are poorly characterized. As part of the Carbon Dioxide and $CH_4$ mission 1.0 (CoMet 1.0) that took place in May – June 2018, we flew a recently developed active AirCore system aboard an unmanned aerial vehicle (UAV) to obtain $CH_4$ and $CO_2$ mole fractions 150-300 m downwind of five individual ventilation shafts in the USCB. In addition, we also measured $\delta^{13}C\text{-}CH_4$, $\delta^2H\text{-}CH_4$, ambient temperature, pressure, relative humidity, surface wind speeds and directions. We have used 34 UAV flights and two different approaches (inverse Gaussian approach and mass balance approach) to quantify the emissions from individual shafts. The quantified emissions were compared to both annual and hourly inventory data, and were used to derive the estimates of $CH_4$ emissions in the USCB. We found a high correlation ($R^2 = 0.7 - 0.9$) between the quantified and hourly inventory data-based shaft-averaged $CH_4$ emissions, which in principle would allow regional estimates of $CH_4$ emissions to be derived by upscaling individual hourly inventory data of all shafts. Currently, such inventory data is available only for the five shafts we quantified though. As an alternative, we have developed three upscaling approaches, i.e., by scaling the E-PRTR annual inventory, the quantified shaft-averaged emission rate, and the shaft-averaged emission rate that are derived from the hourly emission inventory. These estimates are in the range of 325 – 447 kt $CH_4$/year for the





inverse Gaussian approach and 268 – 347 kt $CH_4$/year for the mass balance approach, respectively. This study shows that the UAV-based active AirCore system can be a useful tool to quantify local to regional point source methane emissions.

## 1 Introduction

Methane ($CH_4$) is the second most abundant anthropogenic greenhouse gas (GHG), only second to carbon dioxide ($CO_2$).
Although its abundance is lower than that of $CO_2$, $CH_4$ has a warming potential 28 times greater on a 100-year time frame (Etminan et al., 2016; Van Dingenen et al., 2018). In 2020, its mole fraction reached a global mean of higher than 1870 ppb (Dlugokencky, 2020), a level more than 2.5 times that of preindustrial times. This is mainly attributed to anthropogenic emissions over the last 270 years. Natural $CH_4$ is produced through reservoirs like wetlands and oceans, while anthropogenic $CH_4$ originates from sources like agriculture, waste management, biomass burning, and
exploitation, distribution and use of fossil fuels (Kirschke et al., 2013; Saunois et al., 2016b).

Exploitation of fossil fuels is one of the major contributors of anthropogenic $CH_4$. In the years 2003 to 2017, fossil fuel production and use contributed to an average of 35 % (range 30 – 42 %) of the total annual anthropogenic $CH_4$ emissions, with a mean emission estimate of 128 (range 113 – 154) Tg $CH_4$/year (Saunois et al., 2016a,b 2020).
However, the magnitudes of $CH_4$ emissions are characterized with high uncertainties (Kirschke et al., 2013; Saunois et al., 2017; Turner et al., 2019), with uncertainties of fossil fuel production and use ranging from 20 to 35 % (Saunois et al., 2020). A substantial part of the emitted $CH_4$ from fossil fuel production and use (~33 %, i.e., 41 Tg $CH_4$/year) comes from atmospheric emissions of $CH_4$ from coal mine operations, including underground mining and opencast mining, as well as post-mining activities. Coal mining accounts for ~ 12 % of the total anthropogenic methane emissions worldwide.
When hard coal is extracted by cracking the coal from the bedrock, as well as when the coal is processed via both crushing and pulverization, large quantities of $CH_4$ are released (Zazzeri et al., 2016). The $CH_4$ stored in the coalbed originates from carbonification of biomass (Swolkień, 2020). In the underground mines, part of $CH_4$ is captured via drainage systems and then transported to the surface where it is utilized. The remaining $CH_4$ that has not been captured releases into the mine working area and is then diluted with airflow and vented directly to the atmosphere
through ventilation shafts at the surface to keep the concentration of coal gas within limits for working safety. For many mines the exact amount of $CH_4$ emitted to the atmosphere through these ventilation shafts is poorly characterized and even if data loggers are used to monitor the emissions for reporting to inventories, they lack accuracy and temporal resolution. Without accurate estimates of emissions, it is challenging to develop appropriate mitigation strategies as well as reliable future climate projections.


Stationary towers (Werner et al., 2003; Andrews et al., 2014; Satar et al., 2016) and aircraft measurements (Karion et al., 2013; Krautwurst et al., 2017; Hannun et al., 2020) are commonly used techniques to obtain atmospheric in-situ





measurements, and in recent years the use of unmanned aerial vehicles (UAVs) have also become a key part of the monitoring and measuring of greenhouse gases. In comparison to aircraft, UAVs are easy to maintain, cheap to obtain,

easy to operate, and require less efforts to obtain permits for flying (Villa et al., 2016; Kunz et al., 2020). These UAVs measure and analyze GHGs in a number of different ways; direct in-situ measurement by lightweight sensors (Nathan et al., 2015; Kunz et al., 2020; Martinez et al., 2020; Tuzson et al., 2020), tethered UAV sampling (Turnbullet al., 2014; Brosy et al., 2017; Allen et al., 2019; Shah et al., 2020), and on-board sampling for later analysis (Lowry et al., 2015; Brownlow et al., 2016; Chang et al., 2016; Greatwood et al., 2017; Andersen et al., 2018).


This study is part of the Carbon Dioxide and Methane (CoMet) mission. The overall goal of CoMet is to prepare the future "Merlin mission" (Ehret et al., 2017). In this context, CoMet tries to obtain independent observations of GHG emissions by developing and evaluating new methodologies that can also be used for the validation of satellite measurements (Fix et al., 2018; Swolkień, 2020; Fiehn et al., 2020). Here, in-situ as well as active and passive remote

sensing measurements are used to quantify $CO_2$ and $CH_4$ emissions, which are deployed on different airborne and mobile ground-based platforms. One of the focuses of the CoMet campaign was to quantify the regional $CH_4$ emissions from the Upper Silesian Coal Basin (USCB) (Nickl et al., 2020). The USCB, located in the southern part of Poland, is a region with strong ties to hard coal mining, and is home to more than 70 mining facilities, including coal piles, coal waste heaps, and underground mining networks. According to the European Pollutant Release and Transfer Register (E-

PRTR), the USCB emitted 447 kt $CH_4$ in 2017 (E-PRTR, 2017), with individual coal mine ventilation shafts ranging between emission rates of 0.03 to 20 kt $CH_4$/year. This makes the USCB a strong contributor to the annually emitted $CH_4$ from Europe, being responsible for 27.3 % of the total European $CH_4$ emissions of 1642 kt $CH_4$/year in 2017 according to E-PRTR With the large emission of $CH_4$, and large uncertainties, the USCB is an important region to study and quantify the emitted $CH_4$ from the contributing sources.


Between May 18 and June 1 2018, we performed 59 UAV-based active AirCore flights downwind of individual coal mine ventilation shafts, quantifying the $CO_2$ and $CH_4$ emissions using both an inverse Gaussian approach and a mass balance approach. Isotopic signatures of $\delta^{13}C$-$CH_4$ and $\delta^2H$-$CH_4$ were also obtained by analyzing air samples collected by AirCore during flight. Here we present quantified emissions of 34 active AirCore flights based on atmospheric sampling of

$CO_2$ and $CH_4$ downwind of five individual coal mine ventilation shafts spread across the USCB. These are compared to individual coal mine ventilation shaft inventories, and are then scaled up to estimate the regional USCB $CH_4$ emissions. The upscaled results are compared to regional inventories from E-PRTR (E-PRTR, 2017) as well as previous regional emission estimates from Fiehn et al. (2020) and Kostinek et al. (2021). Isotopic signatures of $\delta^{13}C$-$CH_4$ and $\delta^2H$-$CH_4$ are presented for all five individual coal mine ventilation shafts and compared to previous measurements and known isotopic

signature sources. Section 2 presents the experimental setup that was used as well as the flight data and the methodology to determine emissions. Section 3 contains the results and discussions of the isotopic signatures, the



quantified $CH_4$ emissions and comparisons with annual and hourly inventories, quantified $CO_2$ emissions, and regional USCB emission estimates that are scaled up from quantified shaft ventilation emissions of $CH_4$ and $CO_2$. A conclusion is given in Sect. 4.

## 2 Methodology

### 2.1 UAV-based Active AirCore system

The active AirCore system was introduced in Andersen et al. (2018), and further refined in Andersen et al. (2021). The active AirCore system is an air sampling tool which collects air along the trajectory of a UAV flight by pulling air through a long coiled-up stainless-steel tube. The pump is a small KNF020L micropump, which provides a vacuum downstream of a 45 μm pinhole orifice in order to create conditions for critical flow. Thus, the sampling flow rate of the AirCore only depends on the upstream pressure (ambient pressure), which is measured through the datalogger, along with ambient temperature, ambient relative humidity, temperature within the carbon fibre box housing, and GPS coordinates. This study used three different active AirCore systems, all having 1/8 in. tubing. The lengths of the AirCore were 48.2 m, 46.9 m, and 48.5 m, with estimated volumes of 323 cc, 315 cc, and 325 cc, respectively. The UAV that the active AirCore system is attached to is a DJI Inspire Pro 1. Once an air sample has been obtained, the air is analyzed by a cavity ringdown spectrometer (CRDS, model no. G2401-m, Picarro Inc.) for $CO_2$, $CH_4$, and CO mole fractions. The CRDS used a high-$CH_4$ analysis mode due to the large range of observed $CH_4$ mole fractions (up to 200 ppm). A two-point calibration was used using a known WMO-scale gas mixture around ambient $CH_4$ mole fractions (WMO X2007, X2004A, and X2014A scales for $CO_2$, $CH_4$, and CO, respectively), and a certified mole-fraction gas mixture from the Dutch National Metrology Institute (VSL) containing a high mole-fraction of $CH_4$ (301.1 ppm).

Directly after the CRDS analysis, the AirCore samples were collected in Tedlar bags for further analysis of isotopic signatures of $\delta^{13}C$-$CH_4$ and $\delta^2H$-$CH_4$. The isotopic composition was determined by analyzing the samples stored in the Tedlar bags using a continuous flow isotope ratio mass spectrometer system. More details about the analytical system and the calibration are provided in Brass and Röckmann, 2011; Röckmann et al, 2016; Menoud et al., 2021. Out of the 59 flights performed during this study, the air samples from 34 flights were stored in Tedlar bags for further analysis of isotopic composition. Borynia VI, Pniowek IV, and Pniowek V had two separate days where isotopic compositions were measured, while Brzeszcze IX and Zofiowka IV had 1 day. Each day collected between 4 and 5 samples which were used to determine the isotopic signature using a keeling plot.

### 2.2 Meteorological data

During the first few flights of the campaign, meteorological parameters were measured using a radiosonde (Sparv Embedded AB, Sweden, model S1H2-R) identical to the one used in Andersen et al. (2021). The radiosonde was





tethered through a fishing pole for easier retrieval and reuse, but was lost during the fourth flight due to getting too close to power lines. Four flights had radiosonde profiles to estimate the wind speeds and directions. Flights #5 to #33 were obtained from a nearby meteorological station operated by the Polish meteorological office (IMGW). This was the Katowice Synoptic meteorological station, located at coordinates 50.240556N, 19.032778E. The use of this meteorological data, located a few tens of kilometers away from the measurement sites, may add significant uncertainty to the wind speed and direction for those flights, which was not quantified. For the second half of the campaign, from flight #34 to #59, a mobile onsite meteorological station was used. The surface wind speed and wind direction were measured using a Campbell CSAT3 3-D Sonic Anemometer. The CSAT3 has an operating temperature range of -30 °C to 50 °C. A comparison study of two anemometers, Campbell CSAT3 and Gill R3-50, conducted by Grare et al. (2016) showed that the Campbell CSAT3 measurements are sensitive to small changes in wind direction. The mean differences wind speed and wind direction between the Katowice Synoptic meteorological station and the mobile meteorological stations for flights #34 and onward were $1.7 \pm 0.7$ m/s and $38.8 \pm 29.6°$, respectively.

## 2.3 Flight information

From an internal CoMet inventory based on E-PRTR 2017 emission data, there are 59 ventilation shafts related to hard coal mining operations located within the USCB. Fig. 1 indicates the size of this region. We sampled air from 5 of these ventilation shafts based on their accessibility, and performed a total of 59 flights during the period from May 18 to June 1, 2018. 36 of the 59 flights fulfilled the sampling criteria presented in Andersen et al. (2021). The flights were performed downwind of a specific ventilation shaft while flying perpendicular tracks transecting the plume at incremental heights. This effectively creates a vertical curtain transecting the ventilation shaft plume. The curtain is spaced out into gridded boxes in horizontal($y$)- and vertical ($z$)-direction of size equal to the largest distance between two data point coordinates in the flight, and the largest altitude difference between two point coordinates throughout the flight. The criteria states that the mean wind speed during the flight is larger than 2 m/s and that the flights are performed perpendicular to the wind direction (within 15 degrees). Table (1) shows the number of flights per shaft that fulfilled these criteria, along with the number of measurement days present for each shaft. The flight pattern for the flights was a 'curtain' shape downwind the plume, attempting to intersect the plume at different altitude levels. Fig. 2a shows an example of this pattern. The flight duration varied between 8 and 12 minutes, and distances downwind the plume ranged between 100 to 350 m downwind the ventilation shafts.





**Table 1.** The location of the sampled ventilation shafts, along with the number of days of sampling occurred for each shaft and the number of successful flights each shaft has for emission quantification.

| Coal mining ventilation shaft | Latitude | Longitude | Flights per shaft | Days with sampling |
|---|---|---|---|---|
| Borynia VI | 49.996697°N | 18.648178°E | 4 | 2 |
| Brzeszcze IX | 50.009589°N | 19.156781°E | 5 | 1 |
| Pniowek IV | 49.980367°N | 18.676131°E | 7 | 1 |
| Pniowek V | 49.975407°N | 18.735400°E | 15 | 5 |
| Zofiowka IV | 49.968117°N | 18.627664°E | 5 | 1 |

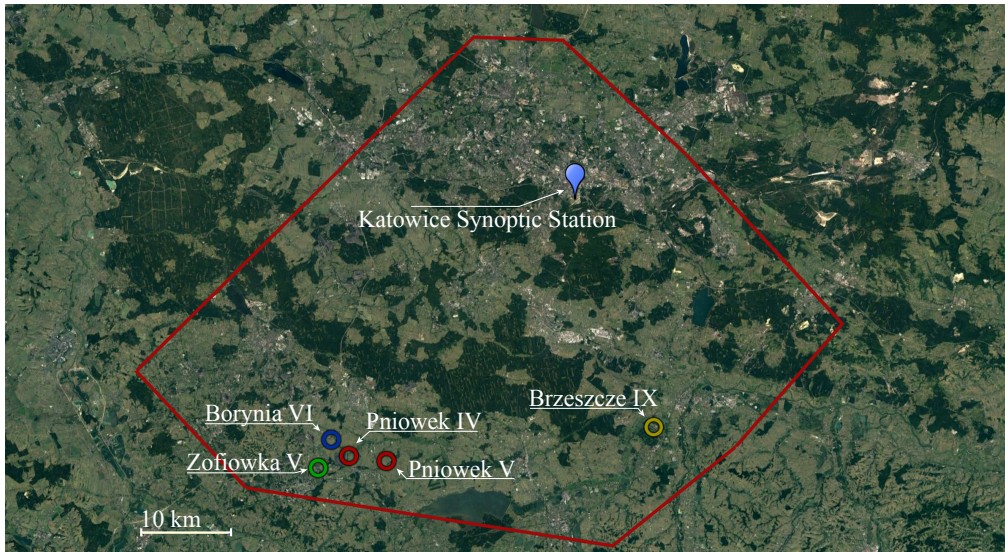

**Figure 1**. Google map showing the location of the 5 measured facilities (round markers) and the meteorological station where wind data for flights #5 to #33 was obtained. The red border indicates the total size of the Silesia Coal Basin where the majority of coal mining shafts were located. We have primarily performed measurements in the south-western part of the region (© Google Maps).

## 2.4 Emission determination

The emitted $CH_4$ emanating from the ventilation shafts is quantified using the methodology derived in Andersen et al.
(2021). At each ventilation shaft, $CH_4$ is vented to the atmosphere through one or more diffusers. Given the distance of 100 – 300 m between the UAV measurements and the ventilation shaft, the emission source can be regarded as a point source. The gridded plane is then used to quantify the emitted emission by applying an inverse Gaussian approach and a mass balance approach. The Gaussian model is given as:





$$C'(x,y,z) = \frac{Q}{2\pi\,\sigma_y\,\sigma_z\,u}\, exp\left(-\frac{1}{2}\left(\frac{y}{\sigma_y}\right)^2\right) \cdot$$

$$\left[exp\left(-\frac{1}{2}\left(\frac{h-z}{\sigma_z}\right)^2\right) + exp\left(-\frac{1}{2}\left(\frac{h+z}{\sigma_z}\right)^2\right)\right] \cdot \frac{V}{M_{CH_4}} \qquad (1)$$


where $C'$ is the dry mole fraction at a given position $x$, $y$, and $z$, which are the projected positional coordinates downwind the plume, across the plume horizontally, and across the plume vertically. The units of $C'$ ($x$, $y$, $z$) in mol/mol, and the units of $x$, $y$, and $z$ are given in m. The emission rate $Q$ is given in kg/s, the wind speed $u$ in m/s, and the stack height $h$ is given in m. The parameters $\sigma_y$ and $\sigma_z$ describe the dispersion of the pollutants in the

horizontal- and vertical direction, respectively, and have units of m. $V$ is the dry molar volume in m$^3$/mol, and $M_{CH_4}$ is the molar mass of CH$_4$, 0.016 kg/mol.

For the mass balance approach, the gridded flight pattern is extrapolated into a full 2D plane using a kriging method, to which the mass balance equation is applied. Fig. 2 shows a measured UAV-based active AirCore profile of CH$_4$ mole

fractions along with the 2D extrapolated kriged CH$_4$ plane, and the inverse Gaussian's estimate plane of CH$_4$ mole fractions. The mass balance equation is given as:

$$Q = \frac{v \cdot \Delta X \cdot M_{CH_4}}{R \cdot T} \sum_{i}^{k_i} \sum_{j}^{k_j} C_{i,j} \cdot P_{i,j} \qquad (2)$$

where the output of the emission rate $Q$ is in kg/s, $v$ is the wind speed in m/s and assumed to be constant throughout the duration of the flights, $k_i$ is the number of horizontal grid boxes in the kriged plane, $k_j$ is the number of vertical grid boxes in the kriged plane, $M_{CH4}$ is the molecular mass of CH$_4$ in kg/mol, $C_{i,j}$ is the CH$_4$ mole fraction in grid box $i$, $j$ in mol/mol, $\Delta X$

is the area of each grid box in m$^2$, $R$ is the universal gas constant, 8.3145 kg m$^2$/s$^2$ K mol, $T$ is the temperature in K, and $P_{i,j}$ is the pressure at each grid box in Pa.

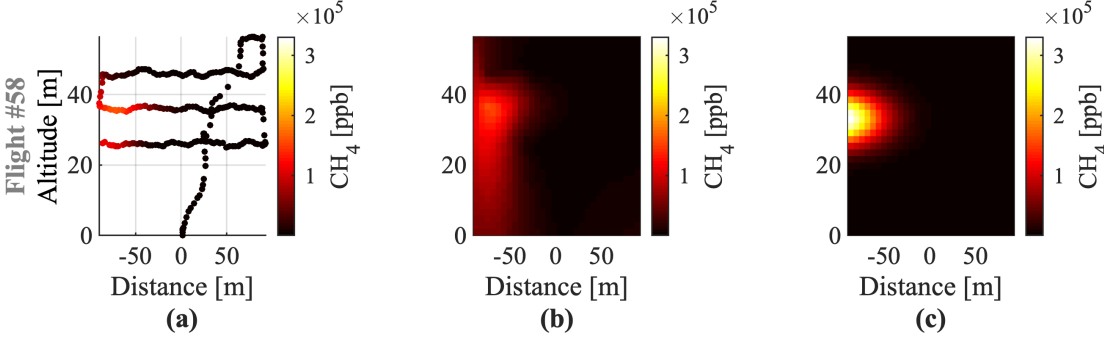

Figure 2. (a) a sampled downwind CH$_4$ mole fraction profile, (b) a kriged extrapolated 2D plane of CH$_4$ mole fractions for the mass balance approach and (c) an estimated 2D CH$_4$ mole fraction plane using the parameters retrieved from the inverse Gaussian approach.





## 2.5 Inventory emissions


The E-PRTR inventory gives the annual emission for each coal mine in the Silesia region. An internal CoMet inventory, which is based on reported 2018 E-PRTR inventories (Gałkowski et al., 2021), lists 59 facilities related to coal mining operations in the USCB, and divides the annual coal mine inventory by geo-localized (via Google Earth) active ventilation shafts for each coal mine. For the comparison used in this study, the active ventilation shafts are assumed to be the same as the ones stated in the internal CoMet inventory, but the E-PRTR values that are being divided equally among active shafts have been updated to the reported E-PRTR 2018 inventories. Pniowek, with a reported emission rate of 54.8 kt $CH_4$/year and three active shafts thus yields an average emission rate of 18.3 kt $CH_4$/year for ventilation shafts Pniowek III, IV, and V. The inventory value for Borynia VI is 7.4 kt $CH_4$/year, for Zofiowka IV 12.7 kt $CH_4$/year, and for Brzeszcze IX 6.9 kt $CH_4$/year.



A second set of inventory data for May to June 2018 is also used for comparison during this study. This is hourly data calculated from raw $CH_4$ concentration measurements and air flow rate measurements obtained within each specific ventilation shaft. Fig. 3 shows a schematic design of a ventilation shaft. The concentration of $CH_4$ is measured with an EMAG-Serwis type DCH methane sensor placed 10 to 15 m down into the exhaust shaft. This sensor has a measurement range of 0 – 100 % with measurement errors of 5 % of the reading value. The conditions are often rough and the relative humidity is high, and the readings of relative humidity could exceed 100% when the filter is wet. The air flow rate is measured using a Prandtl's tube located between the main valve and the fan. According to Swolkień (2020), the relative uncertainty for the air flow rate is 10 %. According to the statements of ventilation engineers, about 5% of the vented air to the atmosphere is from air inflow via the ventilation shaft closure, and we have taken that into account during the calculation of the hourly emission rates, i.e., $CH_4$ concentrations multiplied by 95% of the measured air flow rates.



The conversion into $CH_4$ emissions rate is done as follows:

$$Q_{Inventory} = \frac{P \cdot V_{flow}}{R \cdot T} \rho \qquad (3)$$

Where $P$ is the atmospheric pressure in Pa, $R$ is the universal gas constant in J mol$^{-1}$ K$^{-1}$, $T$ is the ambient temperature in K, $V_{flow}$ is the volumetric flow rate of $CH_4$ in m$^3$ s$^{-1}$, given by the air flow rate multiplied by the $CH_4$ concentration. Lastly, $\rho$ is the molar density of $CH_4$ in g mol$^{-1}$ (16.043 g mol$^{-1}$). A temperature of 20 °C and a pressure of 101325 Pa was used for the calculation.




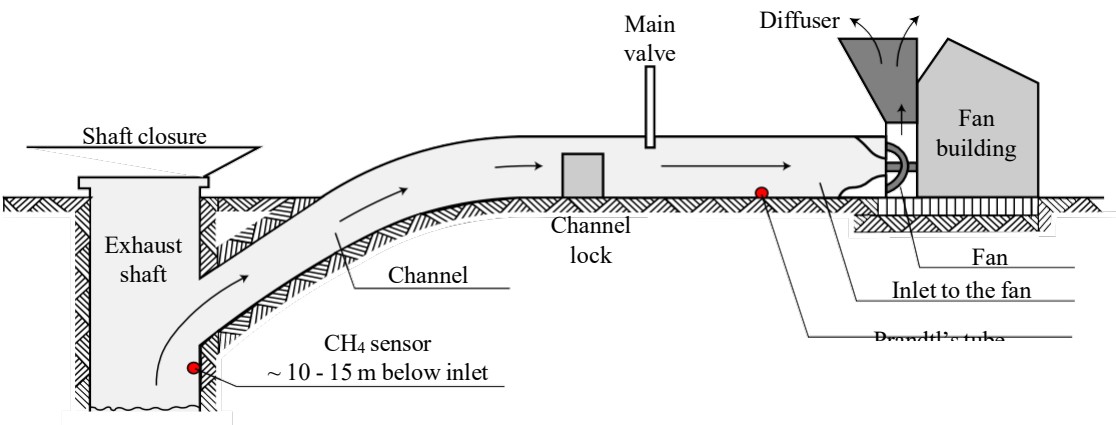

**Figure 3.** Figure from (Swolkień (2020), Fig. 5) showing a coal mine ventilation shaft scheme. This Figure has been re-illustrated with updated graphics and readability for this paper. The original Figure was published under a Creative Commons Attribution 4.0 International License, http://creativecommons.org/licenses/by/4.0/.

## 2.6 Up-scaling

As mentioned in Sect. 2.3, more than 70 facilities related to coal mining operations are located in the USCB. According to the internal CoMet inventory, 59 are active ventilation shafts. After obtaining $CO_2$ and $CH_4$ emissions from 5 of the 59 shafts in the USCB, three distinct approaches are used to obtain an estimate of the regional emission rate. The first method uses the linear correlation of shaft-averaged emissions between our UAV quantified and high frequency (hourly) reported emissions to scale the annual E-PRTR emissions. To avoid the large influence of the intercept, the linear curve has been forced through zero, making the slope the only factor to scale the emissions. The second approach uses the mean quantified shaft emissions, multiplied with the number of ventilation shafts in the region. The third approach scales the mean hourly inventory emission rate to derive the mean quantified emission rate based on the linear correlation of shaft-averaged emissions between our UAV quantified and high frequency (hourly) reported emissions, which is then multiplied by the number of active ventilation shafts in the region.

## 3 Results and discussion

### 3.1 Isotopic signature

Fig. 4 shows the sampled isotopic signatures of $\delta^{13}C$-$CH_4$ and $\delta^2H$-$CH_4$ from the flights during the study, separated into different shafts and different days. For the five sampled ventilation shafts, the $\delta^{13}C$-$CH_4$ values ranged between -53.4 and -41.3 ‰ and the $\delta^2H$-$CH_4$ values ranged between -175.0 and -151.2 ‰. According to Sherwood et al., 2021, isotopic signature values from coal mining vary from country to country and the source signature in Poland was found to be -48 ± 15 (± 1σ) ‰ for $\delta^{13}C$-$CH_4$ and -194 ± 37 for $\delta^2H$-$CH_4$, respectively. All the isotopic signatures found from the UAV active AirCore



flights. Source signatures found during the same measurement campaign, CoMet 1.0, by other groups indicate that the source

signatures for $\delta^{13}$C-CH$_4$ and $\delta^2$H-CH$_4$ in the Upper Silesia Coal Basin range between -59.4 to -41.0 % and -218 to -142 %, respectively (Stanisavljevic, 2021). Overall, the addition of $\delta^{13}$C-CH$_4$ and $\delta^2$H-CH$_4$ measurements, and the good agreement between the found source signatures with those of other groups during the same campaign, indicate that we have clearly sampled the coal mine ventilation shafts using the UAV-based active AirCore system. Based on what is shown in Fig. 4 it is unlikely that other regional CH$_4$ sources (such as biomass burning, landfills, and ruminants) have influenced the active

AirCore measurements.

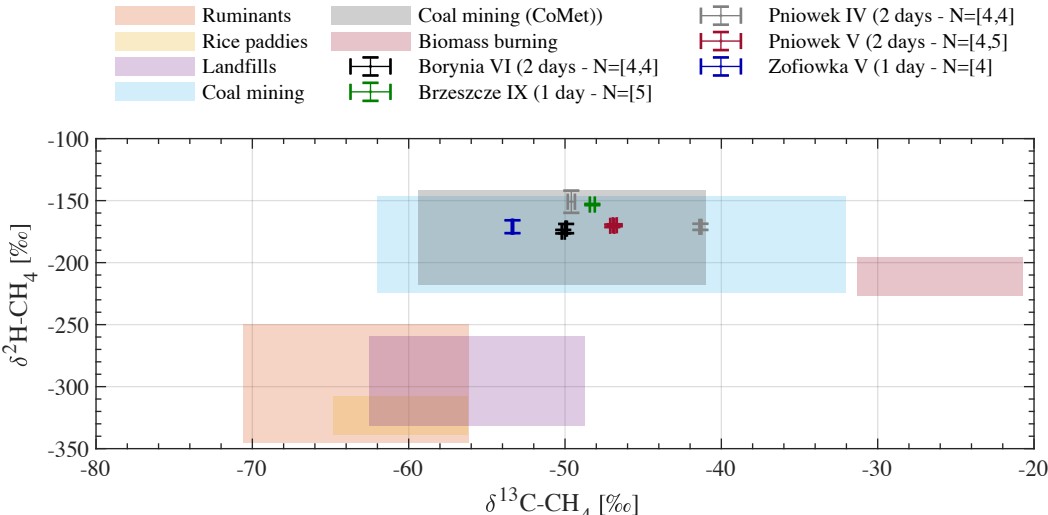

**Figure 4.** Scatter plot indicating the isotopic signature for each measured ventilation shaft. The shaded areas indicate typical $\delta^{13}$C-CH$_4$ and $\delta^2$H-CH$_4$ values for different CH$_4$ sources, and are given with a $1\sigma$ uncertainty. The values and uncertainties for coal mining are determined from measurements in Poland, and for other sources from the whole world (Sherwood et al., 2021; Lan et al., 2021). The gray-

shaded area indicates the isotopic signatures found from other groups during the CoMet 1.0 campaign, and represents the calculated weighted average for the coal in the USCB (Stanisavljevic, 2021; Menoud et al., 2020)

### 3.2 Quantified CH$_4$ emissions

Fig. 5&6 show the estimated CH$_4$ emission rates from individual ventilation shafts, for each day. Averages range between $2.7 \pm 2.0$ and $15.0 \pm 2.3$ kt/year for the inverse Gaussian approach, and between $0.8 \pm 1.0$ and $14.4 \pm$

3.7 kt/year for the mass balance approach. Large variations are seen from day-to-day for the same coal mine ventilation shafts. The inverse Gaussian approach and mass balance approach have a mean difference of 2.5 kt/year, with a maximum difference of 8.9 kt/year on May 31. This is likely due to the majority of the plume being located outside of the gridded curtain, which causes the inverse Gaussian to move the center line of the plume off the grid to obtain the best fit between model and data, while the mass balance is constrained to only include what is included in the kriged plane. The same is

seen in the first flight on May 25 for Pniowek IV (see Fig. 6), where the majority of the inverse Gaussian plume is located





outside the measured grid.

Three of the days were either weekend days or holidays. May 19 was a Saturday, while May 20 and May 28 were public
holidays in Poland. The emission rates of $CH_4$ could have been affected by irregular mining activity on these particular

days. If mining operation were reduced on those days, less coal would have been cracked from the bedrock, and would
lead to less $CH_4$ venting to the atmosphere, which will be further discussed in Sect. 3.3. Pniowek V was sampled on two
of these days and can be compared to normal days. The holidays have an average estimate of 7.6 ± 3.6 kt/year for the
inverse Gaussian, whereas the average during the sampled weekdays is 12.1 ± 2.7 kt/year. For the mass balance
approach the mean weekend/holiday emissions are 8.3 ± 2.7 kt/year, while the weekday emissions have an average of 9.2

± 7.4 kt/year, so here the difference is not significant.   May 31 only has one successful flight, and only has mole
fraction enhancement along the edge of the flight (see supplement Fig. 15 flight #56), which leads to underestimation of the
emission rate using the mass balance approach. Comparatively, the inverse Gaussian finds the plume center outside of the
sampled plane, and estimates a much larger emission rate. Excluding the flight on May 31, the weekly mean becomes
13.3 ± 2.5 kt/year for the inverse Gaussian and 13.5 ± 1.4 kt/year for the mass balance approach. The weekend/holiday

emissions are for the inverse Gaussian within the range of the error, while the mass balance does not overlap. The ratio
of weekend/holiday emissions to weekday emissions is 0.63 for the inverse Gaussian approach and 0.90 for the mass
balance approach. This may indicate that there is an influence on the emitted $CH_4$ during weekends/holidays. This means
that the quantified emissions of the one day of measuring Brzeszcze IX may also be lower than on normal weekdays.



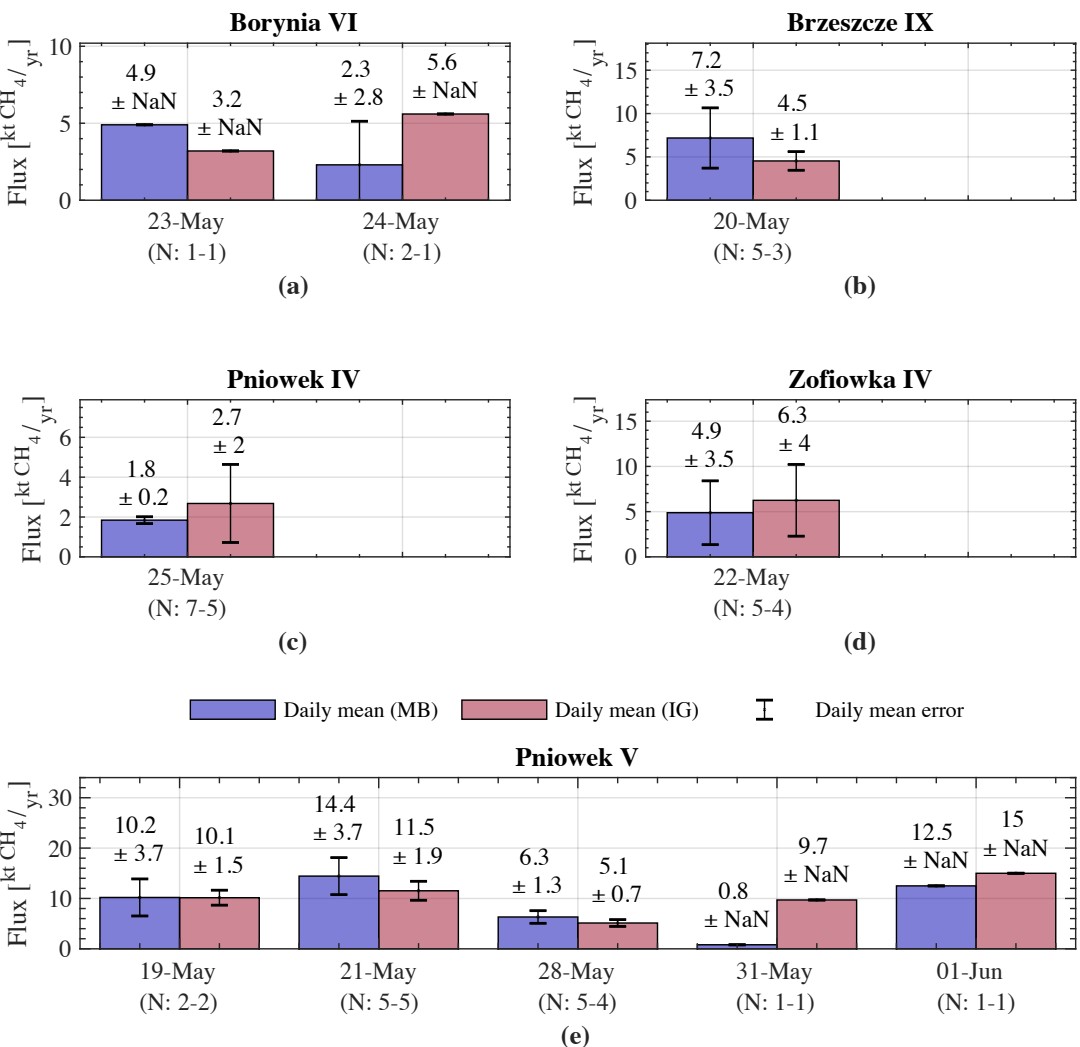

**Figure 5.** CH$_4$ emission estimates for each ventilation shaft per measurement day. light red: inverse Gaussian approach; light blue: mass balance approach. The bar height is the average of all flights during a specific day. Error bar indicates the standard deviation of the individual flights for that specific day, where the number of flights used for each bar is indicated with N. The two values for N refer to the mass balance approach and inverse Gaussian approach, respectively. The error is indicated as NaN when only one estimate is available.

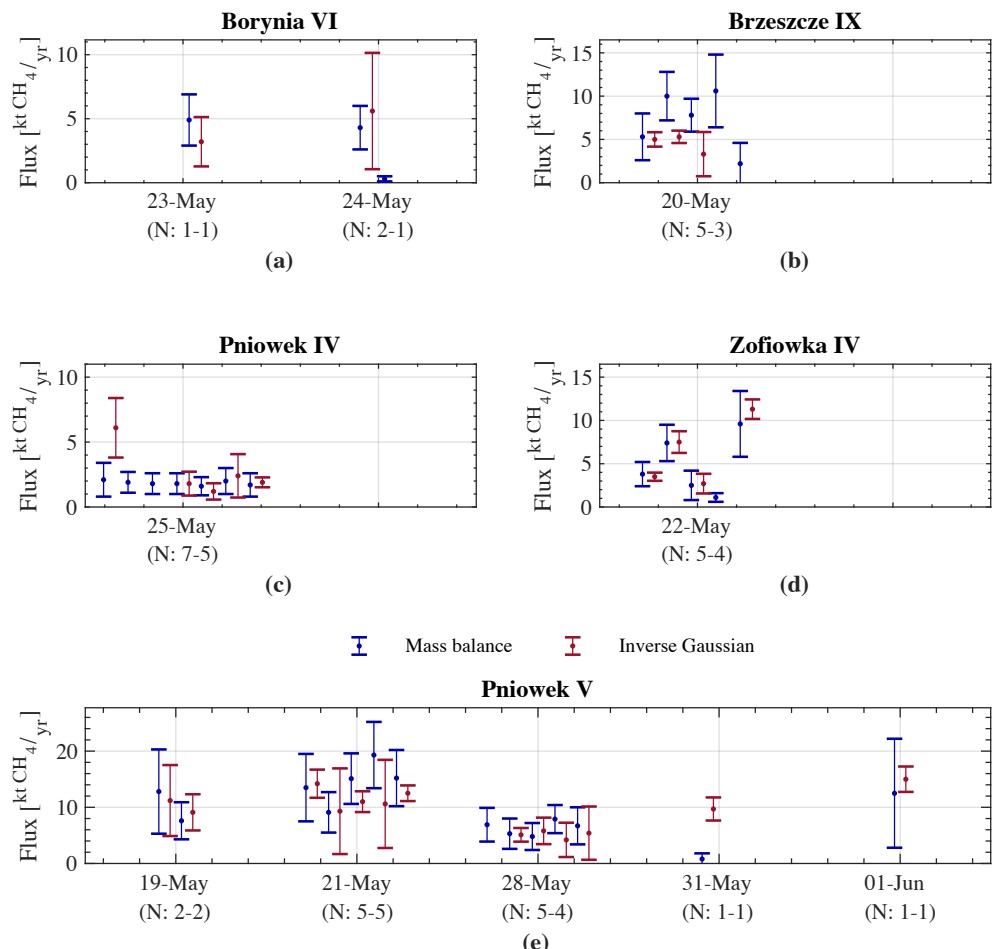

**Figure 6.** The quantified CH$_4$ emission for each flight divided into different ventilation shaft and separated by individual flight days. The emissions are also color differentiated by inverse Gaussian approach (red) or mass balance approach (blue).

### 3.3 Comparison with inventory

Fig. 7 shows the hourly inventory emissions for each ventilation shaft. The inventory reported to the E-PRTR is based on this data. Note that inventory measurement for Borynia VI is missing for the period between May 19 and May 30 (Fig. 7a). We assume this was due to a malfunctioning CH$_4$ sensor inside the ventilation shaft. The listed inventory data for Borynia VI in Table (2) was therefore calculated with data from May 30 to June 02. The Borynia VI inventory may therefore not represent the actual inventory of the days of measurements. The same can be concluded for Brzeszcze IX

(Fig. 7b), which only has one given measurement point. The variability in the emitted emission is clearly seen in the data from Pniowek IV, Pniowek V, and Zofiowka IV (Fig. 7c,d,e).





The gray-shaded areas in Fig. 7 indicate days that were either weekend days or public holidays, and the highlighted red areas indicate flight days. As seen in Fig. 7e, some of the largest emissions occur during weekend/holidays, while

some of the lowest emissions occur during the weekdays. There does not seem to be a consistent difference in emitted $CH_4$ between weekdays and weekend days/holidays, as previously postulated in Sect. 3.2. The $CH_4$ emissions of individual ventilation shafts show large variations, both hour-to-hour and day-to-day.

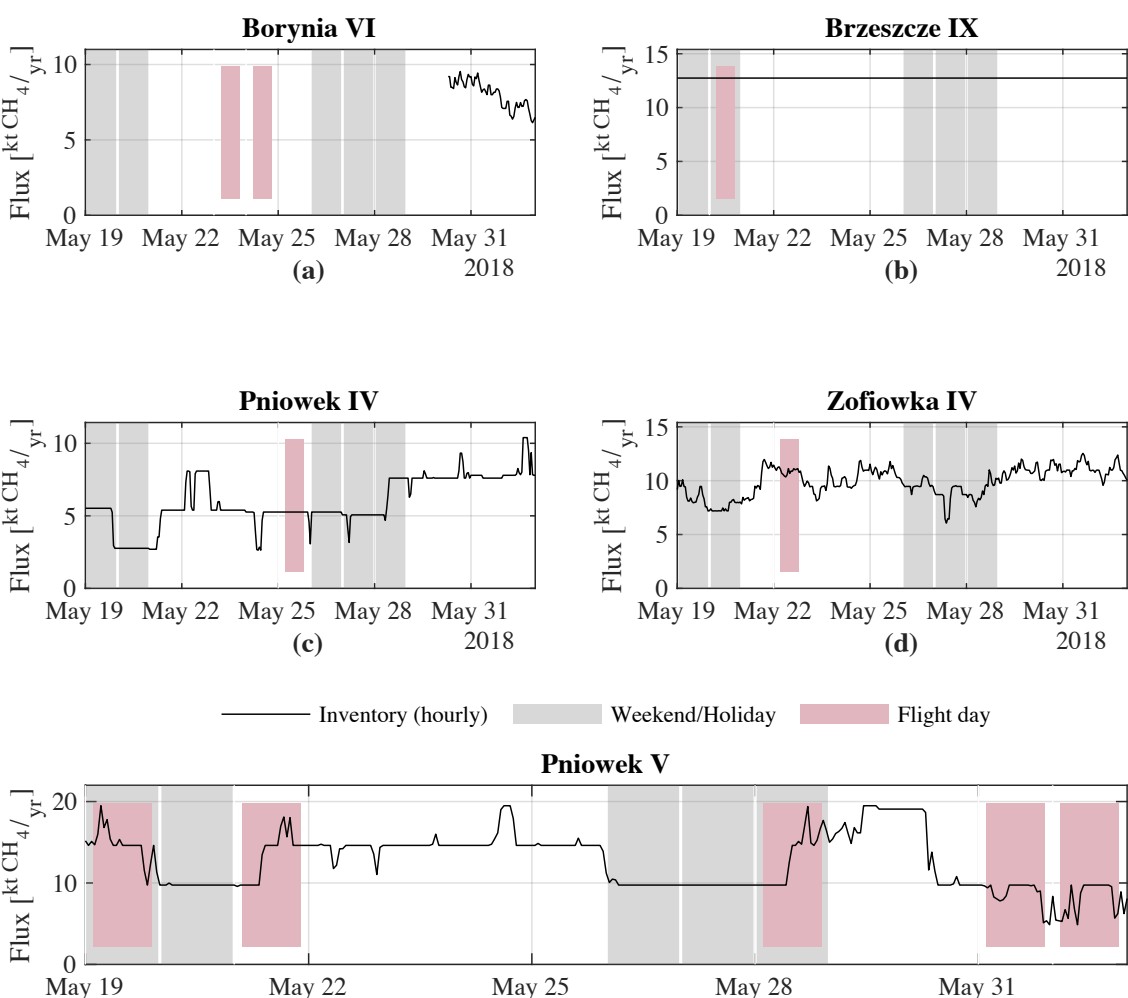

**Figure 7.** Time series of hourly inventory emissions from $CH_4$ concentration and air flow measurements in the shaft for each investigated

coal mine ventilation shaft. The shaded gray areas indicate weekend or holidays, and the shaded red areas indicate days of UAV-based active AirCore sampling. Prior to May 30 data in (a) are missing. In (b) only a constant value is available from May 19 to June 1.

In comparing the quantified $CH_4$ emission rate on an individual flight basis with the annual emission rate reported to the E-PRTR, we found that the correlation is very low ($R^2 < 0.05$). Fig. 8a shows the correlation between the E-PRTR annual





emissions that has been divided by the number of active ventilation shafts for a particular coal mine, and the UAV-based
active AirCore inverse Gaussian quantified $CH_4$ emissions averaged by shaft emissions. Also, here the correlation is low
($R^2 < 0.07$, N = 5). When the total reported mine emissions for a specific mine from the E-PRTR inventory are divided
equally by the number of active shafts, shaft-specific emission info is lost. The non-existing correlation indicates that the
agreement between the snapshot flight quantified emissions with the E-PRTR inventory is poor.


**Table 2.** The statistics for the annual $CH_4$ inventory (E-PRTR (2018)), the hourly inventory during the days of flying, and the UAV-based active AirCore inverse Gaussian quantified $CH_4$ emissions for each coal mine ventilation shaft.

| Shaft | Annual E-PRTR inventory [kt/year] | Hourly inventory [kt/year] | | | | Inverse Gaussian [kt/year] | | | | Mass balance [kt/year] | | | |
|---|---|---|---|---|---|---|---|---|---|---|---|---|---|
| | | N | Min | Mean | Max | N | Min | Mean | Max | N | Min | Mean | Max |
| Pniowek IV | 18.3 | 24 | 2.8 | 6.1 ± 1.6 | 8.4 | 5 | 1.2 | 2.7 ± 2.0 | 6.1 | 7 | 1.6 | 1.8 ± 0.2 | 2.1 |
| Pniowek V | 18.3 | 120 | 8 | 12.3 ± 3.1 | 17.8 | 13 | 4.2 | 9.5 ± 3.5 | 15 | 13 | 0.8 | 9.8 ± 5.0 | 19.3 |
| Borynia VI | 7.4 | 66 | 6.1 | 8 ± 0.9 | 9.5 | 2 | 3.2 | 4.4 ± 1.7 | 5.6 | 3 | 0.3 | 3.2 ± 2.5 | 4.9 |
| Zofiowka IV | 12.7 | 24 | 7.4 | 9.9 ± 1.1 | 11.3 | 5 | 2.7 | 6.3 ± 4.0 | 11.3 | 5 | 1.1 | 4.9 ± 3.5 | 9.6 |
| Brzeszcze IX | 6.9 | 1 | 12.7 | 12.7 ± - | 12.7 | 4 | 3.3 | 4.5 ± 1.1 | 5.3 | 5 | 2.2 | 7.2 ± 3.5 | 10.6 |
| Average | 11.3 ± 5.3 | | 7.4 | 9.8 ± 2.8 | 11.9 | | 2.9 | 5.5 ± 2.6 | 8.7 | | 1.2 | 5.4 3.2 | 9.3 |

The hourly inventory data shown in Fig. 8b is therefore required for a direct comparison with the quantified emissions.
Comparing this data on a daily-averaged basis with daily-averaged flight data sees a slight improvement in the obtained
correlation ($R^2 = 0.23$, N = 9), although the correlation is still weak. Due to the lack of hourly data for Brzeszcze IX, it has
been omitted for the comparison. There can still be large variations on hourly basis, and thus a direct comparison
between the hourly inventory over a day with snapshot flight profiles during the same day may not always align.
Therefore, we have averaged the days together and compare shaft-specific averaged hourly data with shaft-specific averaged
UAV quantified emissions from the same days. This is shown in Fig. 8c, which obtains a stronger correlation than the
two previous comparisons, with an $R^2 = 0.86$ (N = 4). When the linear fit is forced through zero, a higher $R^2$ value (0.95)
is obtained. The quantified emissions are roughly 40 % lower than those of the hourly inventory; however, this is not
significantly when considering the large standard deviation of the measurements.

The much-improved correlation from comparing hourly inventory data from individual shafts as opposed to a total mine
emission divided equally over active shafts (i.e., based on the E-PRTR 2018 inventory), indicates that translating shaft-





quantified snapshot emissions to annual inventories is difficult. The hourly inventory data is not always available, but our evaluation indicate that they are required to make meaningful comparisons between quantified emissions and inventories. Due to the good correlation between the hourly inventory and the quantified emissions per shaft, we can
use the hourly inventory data to scale up the quantified emissions. We use the slope of 1.1 and the intercept of -4.0 of the linear fit to scale up our quantified emissions. This will be discussed in Sect. 4. For the mass balance approach (data not shown), the correlations are also much improved when hourly inventory data is used for comparison, although the $R^2$ values are slightly lower than those for the inverse Gaussian approach.

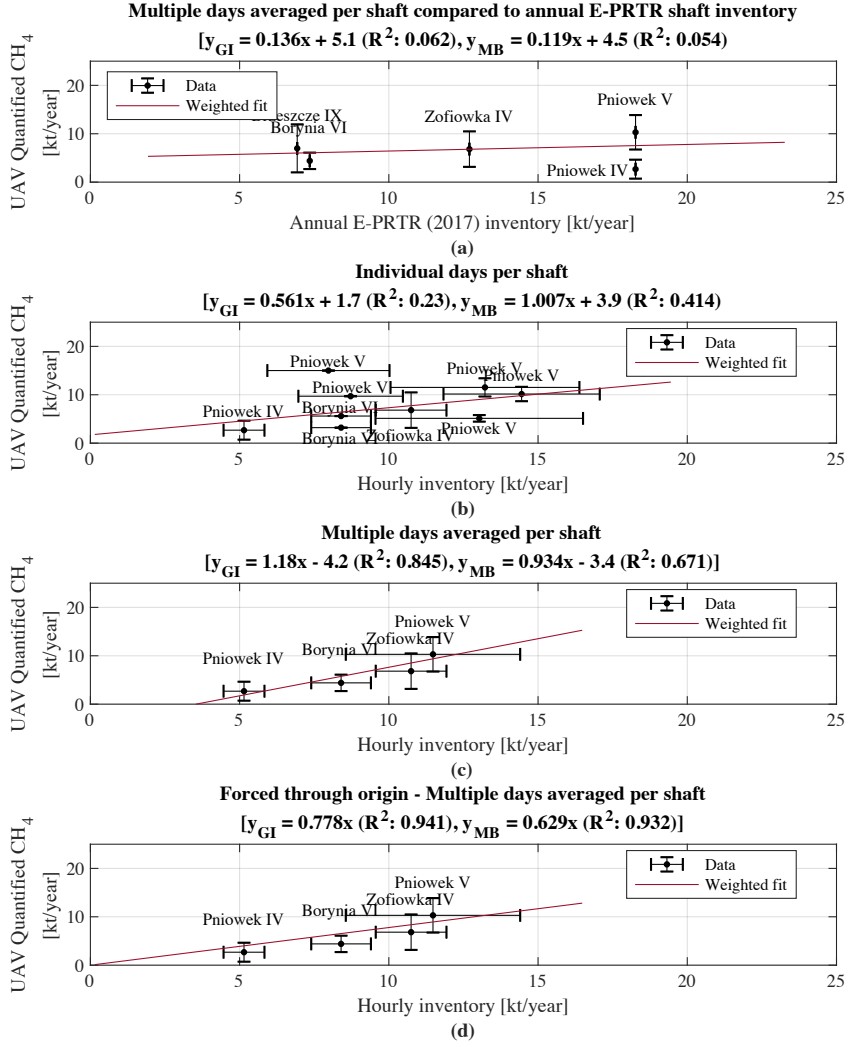

**Figure 8.** Scatter plot of UAV quantified shaft-averaged emissions over multiple days or individual days against annual or hourly inventory data (a) shaft-averaged emissions over multiple days vs. annual coal mine emissions from the E-PRTR 2018 (Gałkowski, 2021) inventory; (b) daily shaft-averaged emissions vs. daily high frequency (hourly) shaft-averaged emissions; (c) shaft-averaged emissions over multiple days vs. shaft-averaged high frequency (hourly) emissions over the same days; (d) same as (c) except that the fit



has been forced through origin. All panels display only the data from the inverse Gaussian approach; however, the title lists the curve fit
from the mass balance approach as well. The E-PRTR inventory has been divided by the number of active ventilation shafts, and the
number of active shafts is taken from the internal CoMet inventory, which had emission profiles based on 2018

Fig. 9 shows the boxplot comparison between estimated emissions from both the inverse Gaussian approach and the mass
balance approach, against the hourly inventory for each ventilation shaft. The inventory data includes data for the same
days as the flights, except for Borynia VI and Brzeszcze XI. As previously mentioned, Brzeszcze XI contains only an
annual estimate, while for Borynia VI inventory data are missing for the specific days when this shaft was sampled.
Pniowek V, the shaft with the best statistics (N = 13 for the inverse Gaussian and N = 14 for the mass balance approach
over 5 different days), has largely overlapping distributions with the hourly inventory data, although leaning towards the
lower end of the hourly inventory distribution. This indicates that this statistical pool is sufficient to accurately quantify
comparable emissions. Pniowek IV and Zofiowka IV both have N = 5 for the inverse Gaussian, and N = 7 and N = 5 for
the mass balance, respectively. Zofiowka IV has overlapping distributions with the hourly inventory, but the quantified
emissions largely span the lower hourly inventory distribution. This is seen with all other shafts as well. Pniowek IV has
only a small overlap with the hourly inventory distribution for both the inverse Gaussian and mass balance approach. This
could be due to variable winds making quantification difficult flights, or perhaps that the flights were performed at times of low
emission that the hourly inventory did not pick up. Brzeszcze IX is difficult to compare, due to the lack of hourly
inventory data, and the only hour inventory data matches the upper end of the inverse Gaussian estimates. Finally, Borynia
VI has the lowest statistics with N = 2 for the inverse Gaussian and N = 3 for the mass balance approach over two
different days. There is no overlap between the distributions. Borynia VI, as well as Brzeszcze IX, are difficult to compare,
due the lack of direct hourly inventory data around the days of flying.

Thus, the measured distributions for Pniowek V, Pniowek IV, and Zofiowka IV all over with the hourly inventory
distributions for the same day, with a minimum of N > 5 flights. The largest overlap is as mentioned found in Pniowek V,
containing several days of sampling and N > 13. These distribution comparisons suggest that although single flight estimates
may not be correlated well with the hourly inventory, the averaged estimates of multiple flights show a strong correlation
with those of the inventory, which suggests that more than one flights are required to obtain a good estimate. Note that for
all shafts, the UAV estimated emission distribution is located on the lower end of the inventory distribution.



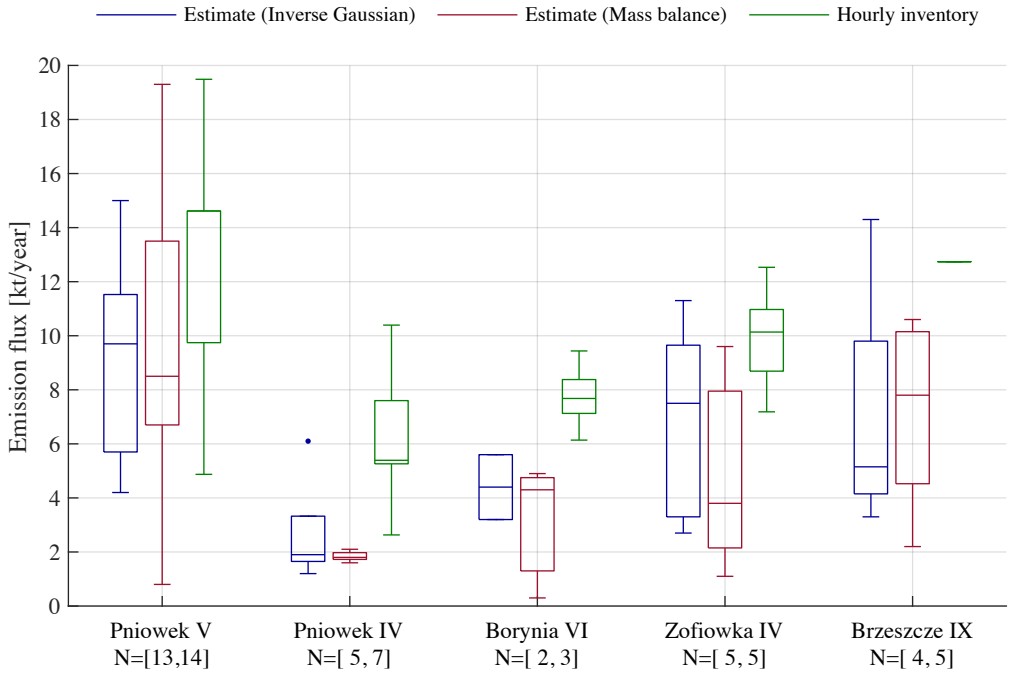

**Figure 9.** Boxplot comparison of estimated emission vs. hourly inventory data. The hourly inventory data has been calculated from shaft emission data from the mining companies, using $CH_4$ concentration and flow rate measurements.

### 3.4 Carbon dioxide emission

Similar to the coal mining shaft sampled in Andersen et al. (2021), a strong correlation is found between the emitted $CO_2$ and $CH_4$. The way of obtaining the emitted $CO_2$ emission using the correlation between $CO_2$ and $CH_4$ mole fractions, the emitted $CH_4$ emissions, and the molar mass constants of $CO_2$ and $CH_4$ is given as:

$$Q_{CO_2} = \frac{Q_{CH_4} \cdot M_{CO_2}}{slope \cdot M_{CH_4}} \tag{4}$$

where $Q_{CH4}$ is the quantified $CH_4$ emission, the *slope* is the slope of the linear fit between $CO_2$ and $CH_4$, and $M_{CO2}$ and $M_{CH4}$ are the molar masses of $CO_2$ and $CH_4$, respectively. There were some flights that had no, or low correlation, and

were thus omitted from the $CO_2$ emission calculation. These were flights with $R^2 < 0.5$. Of the 36 flights that fulfilled the criteria list, the number of flights above an $R^2$ value of 0.5 was 25, with an average $R^2$ of 0.8. The average $CH_4/CO_2$ slope was $4.6 \pm 2.9$ ppm$_{CH4}$ /ppm$_{CO2}$.

Fig. 10 shows the calculated $CO_2$ emission on a daily-averaged basis for each coal mine ventilation shaft. Expectedly, the $CO_2$

follows the same trend as the $CH_4$, seeing strong variations on a day-to-day basis. The mean difference between the inverse





Gaussian and the mass balance approach is 1.5 kt/year. The average $CO_2$ emission rate over all shafts calculated using the inverse Gaussian approach is 4.4 ± 2.2 kt/year, with a minimum of 0.8 ± NaN kt/year and a maximum of 7.2 ± 4.1 kt/year. For the mass balance approach, the average $CO_2$ emission rate is 3.8 ± 2.3 kt/year, with a minimum of 0.5± NaN kt/year and a maximum of 7.5 ± 1.8 kt/year.

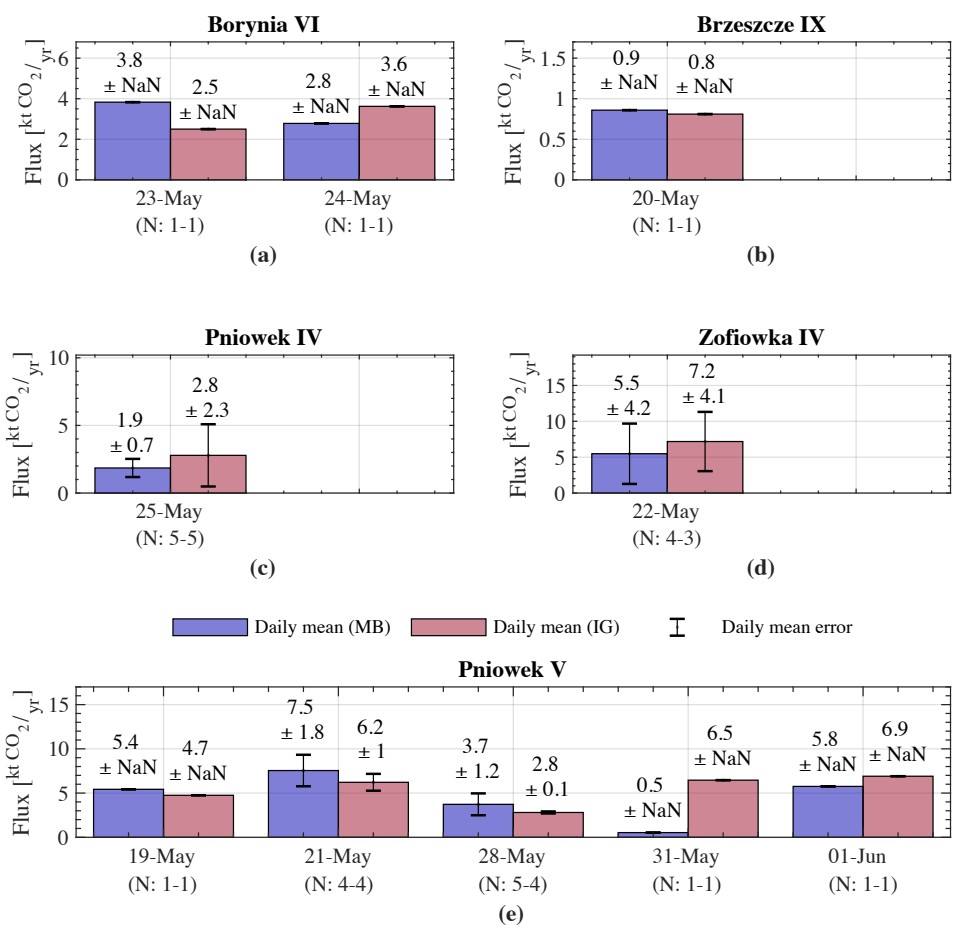

**Figure 10.** shows $CO_2$ emission histograms for each ventilation shaft divided into separate days. Emission quantifications for both the inverse Gaussian approach (light red) and mass balance approach (light blue) are shown. The bar height is the mean of all flights during a specific day.

### 3.4 Upscaling to regional estimates

As shown in Table (2), the mean quantified $CH_4$ emission of the five sampled coal mine ventilation shafts is 5.5 ± 2.6 kt $CH_4$/year for the inverse Gaussian approach and 5.4 ± 3.2 kt $CH_4$/year for the mass balance approach, respectively. For $CO_2$, the mean emission is 4.2 ± 2.2 kt $CO_2$/year for the inverse Gaussian approach and 3.8 ± 2.3 kt $CO_2$/year for the mass balance, respectively. As much as 59 active ventilation shafts are located across the entire USCB. According to the 2018 E-



PRTR inventory, the regional $CH_4$ emissions adds up to 447.9 kt $CH_4$/year, while the regional $CO_2$ emissions are stated to be
35.3 [Mt $CO_2$/year].

Three distinct approaches have been used to obtain an estimate of the regional emission rate. The first method uses the linear
correlation of shaft-averaged emissions between our UAV quantified and high frequency (hourly) reported emissions shown
in Fig. 8d to scale the annual E-PRTR emissions. To avoid the large influence of the intercept, the linear curve has been
forced through zero, making the slope the only factor to scale the emissions. For the inverse Gaussian approach, the slope is
0.744, which multiplied with the 447.9 kt $CH_4$/year inventory results in 332.6 kt $CH_4$/year. For the mass balance, with a
slope of 0.6, the resulting emissions are 268.2 kt $CH_4$/year. These results are shown in Fig. 11a as yellow bars.

The second approach uses the mean quantified shaft emissions of 5.5 ± 2.6 kt $CH_4$/year for the inverse Gaussian approach
and 5.4 ± 3.2 kt $CH_4$/year for the mass balance approach, multiplied with the number of ventilation shafts in the region. This
amounts to a regional emission of 324.5 ± 147.5 kt $CH_4$/year for the inverse Gaussian approach and 318.6 ± 188.8 kt
$CH_4$/year for the mass balance approach, respectively. These emission estimates compare well with the ones from the
previous approach, but are lower than the emissions estimated by Fiehn et al. (2020) and Kostinek et al. (2021). These are
shown in Fig. 11a as blue bars.


The third approach uses the linear curve from Fig. 8c to scale the mean hourly emission rate calculated from hourly
inventory data, to derive the mean quantified emission rate, which is then multiplied by the number of active ventilation
shafts in the region. Here, both the slope and intercept are used for the scaling. The mean hourly inventory emission rate is
10.4 ± 3.1 kt $CH_4$/year. The linear curve using the inverse Gaussian approach has a slope of 1.113 and an intercept of -4.0,
resulting in a derived mean quantified emission rate of 7.6 ± 2.3 kt $CH_4$/year. For the mass balance, a slope of 0.873 and an
intercept of -3.2 results in a derived mean quantified emission rate of 5.9 ± 1.8 kt $CH_4$/year. Multiplying these numbers with
the number of active ventilation shafts results in regional emission rates of 446.9 ± 133.2 kt $CH_4$/year for the inverse
Gaussian and 346.9 ± 103.4 kt $CH_4$/year for the mass balance approach, respectively. The regional estimates for the inverse
Gaussian approach and mass balance approach resulting from the third upscaling approach are shown in Fig. 11a as purple
bars.





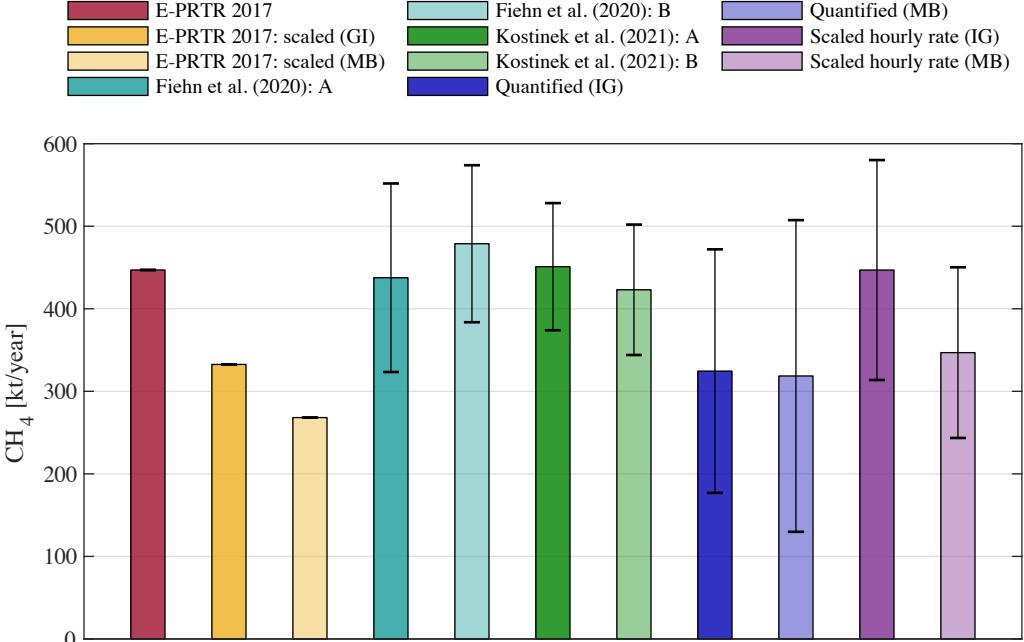

**Figure 11.** A comparison of regional inventory emissions for CH4. The first bar (red) represents the E-PRTR inventory. The second bar (yellow) represents the E-PRTR inventory scaled by the linear fit. Bars three and four (teal) represent the estimated regional emissions from Fiehn et al. (2020) from their two flights. Bars five and six (green) represent the estimated regional emissions from the two flights of Kostinek et al. (2021). Bars number seven (blue) and eight (light blue) represent the regional emission using the quantified inverse Gaussian and mass balance estimates, respectively. The last two bars, ten (purple) and eleven (light purple), represent the scaled regional emission using the inverse Gaussian approach and the mass balance approach, respectively.

Comparing the inverse Gaussian-derived regional emission with both the annual E-PRTR inventory and the regional estimates from Fiehn et al. (2020), the results are close to one another, and are not statistically different when their uncertainties are considered. Fiehn et al. (2020) estimated the regional emissions over two separate flights during the same CoMet campaign to be $437.6 \pm 114.2$ kt $CH_4$/year and $478.8 \pm 95.1$ kt $CH_4$/year, similar to the $447.9$ kt $CH_4$/year E-PRTR inventory. Kostink et al. (2021) also estimated the regional emissions over two separate flights, and found emissions rates of $451 \pm 77$ kt $CH_4$/year and $423 \pm 79$ kt $CH_4$/year. Our estimated emissions appear to be lower, except for the inverse Gaussian-derived scaled hourly rate. Since we have only quantified 5 individual shafts out of 59 active shafts in the region, the small number of quantified shafts could be one of the main causes of the difference.

The upscaling process for $CO_2$ cannot be explored by the same approaches as for $CH_4$, since the linear curves from Fig. 8 are only valid for $CH_4$. Therefore, only the second approach can be used, where the mean quantified $CO_2$ emission will be multiplied with the number of active ventilation shafts in the region. The mean $CO_2$ emission is $4.2 \pm 2.2$ kt $CO_2$/year for the





inverse Gaussian approach and $3.8 \pm 2.3$ kt $CO_2$/year for the mass balance, which yields a regional emission estimate of $0.3 \pm 0.1$ Mt $CO_2$/year for the inverse Gaussian approach and $0.2 \pm 0.1$ Mt $CO_2$/year for the mass balance approach, respectively. This is significantly less than the E-PRTR inventory of 35.3 Mt $CO_2$/year and the estimated regional emissions rates from Fiehn et al. (2020) of $38.2 \pm 22.7$ Mt $CO_2$/year and $35.3 \pm 11.7$ Mt $CO_2$/year. Comparatively, these estimates are

~ 1 % or less of the listed E-PRTR inventory. According to the E-PRTR (2018) inventory, 98.2 % of emitted $CH_4$ in the USCB originates from underground and related operations, 1.5 % coming from opencast mining and quarrying, and 0.3% from waste and waste water management. For $CO_2$, the major contributors are thermal power stations and other combustion installations and production and processing of metals. These account for 78.9 % and 16.3 %, respectively. Residential heating accounts for 2.6 %, while other industrial manufacturing accounts for 2.2 %. However, $CO_2$ emissions from coal

mining activities are not included in the E-PRTR inventory.

The upscaling uses daily snapshots to estimate an annual emission by multiplying the annual average of the five sampled shafts by the number of ventilation shafts in the region. As shown in Sect. 3.3, each ventilation shaft can have significant variations in its daily emissions, thus this adds uncertainty to the daily snapshots extrapolated to an annual emission.

Ventilation shafts can have significantly different emission rates, thus grouping the 5 shafts together to obtain the average does not accurately represent the emission distribution in the whole region. This adds additional uncertainty to the upscaled regional emission. Despite this, we see a good agreement with the two flights from Fiehn et al. (2020), Kostinek et al. (2021) and the E-PRTR inventory for $CH_4$ within the error bars (see Fig. 11a), especially using the third approach of deriving the quantified emissions from hourly inventory data and scaling this to a regional emission rate. This indicates that

the upscaling of the ventilation shafts emission estimated from the UAV-based active AirCore can be a useful tool for relatively cheap and easy-to-obtain regional emission estimates. Estimated regional $CO_2$ emissions are vastly smaller than the suggested regional inventory and also the regional emissions found by Fiehn et al. (2020). The estimated regional $CO_2$ emissions account for ~1 % of inventory, confirming that the coal mine ventilation shafts are not a major source of $CO_2$ in the USCB. This is also reflected in the E-PRTR inventory, which does not list coal mining as a $CO_2$ source at all. Due

to the omission of $CO_2$ emitted from underground coal mining in the E-PRTR inventory, we conclude that the $CO_2$ inventory is missing a source of roughly 1 %.

## 5 Conclusions

It is important to obtain independent estimates of the emission magnitudes from coal mining shafts and verify reported emission inventories to be able to reduce the overall emissions. Using the UAV-based active AirCore system, we have made

atmospheric measurements of $CH_4$ and $CO_2$ mole fractions downwind of five different coal mine ventilation shafts in the USCB. We apply an inverse Gaussian approach as well as a mass balance approach to quantify the $CH_4$ and $CO_2$ point-source emissions for the five sampled ventilation shafts, and compare these estimates with reported inventory data. The





estimated point sources are used to extrapolate a total USCB regional $CH_4$ and $CO_2$ estimate.

The $CH_4$ emission estimates indicate that the coal mine ventilation shafts have highly variable emission rates. Over the five quantified shafts, the quantified emissions using the inverse Gaussian approach range between 1.2 and 15.0 kt $CH_4$/year, with a mean of 5.5 ± 2.6 kt $CH_4$/year. For the mass balance approach, the quantified emissions range between 0.3 and 19.3 kt $CH_4$/year with a mean value of 5.4 ± 3.2 kt $CH_4$/year. This large variability is reflected in the hourly inventory data for the same coal mine ventilation shafts, and it is therefore clear that comparisons of the UAV-based active AirCore 490 quantified emissions and annually averaged inventories show very low ($R^2$ = 0.06). Day-by-day comparisons of the quantified emissions with hourly inventory during the same days yields a better correlation ($R^2$ = 0.23), but the best correlation is found on shaft-by-shaft comparisons, obtaining an $R^2$ of 0.85 for the inverse Gaussian approach and 0.67 for the mass balance approach. Distribution comparisons between the hourly inventory and the quantified emissions show that more flights are beneficial to accurately estimate the shaft emissions. Due to the large variability of 495 the shaft emissions, single flights may sample at times of small or large emission. Correlation between $CH_4$ and $CO_2$ mole fractions is large for all flights (average $R^2$ = 0.8) and has an average slope value of 4.6 $ppm_{CH4}$ /$ppm_{CO2}$. Quantified $CO_2$ emissions for the combined five ventilation shafts yield an average of 4.4 ± 2.2 kt $CO_2$/year for the inverse Gaussian and 3.8 ± 2.3 kt $CO_2$/year for the mass balance approach.

To obtain regional estimates, we have used three upscaling approaches by scaling the E-PRTR annual inventory, the quantified shaft-averaged emission rate, and the shaft-averaged emission rate that are derived from the hourly emission inventory. The first approach obtains emission rates of 333 kt $CH_4$/year from the inverted Gaussian approach and 268 kt $CH_4$/year from the mass balance approach, respectively, which compares well with the second approach of 325 ± 148 kt $CH_4$/year (Gaussian) and 318.6 ± 189 kt $CH_4$/year (mass balance). These estimates are slightly lower than the previous 505 results from Fiehn et al. (2020), Kostinek et al. (2021) and the E-PRTR inventory of 448 kt $CH_4$/year. The third approach results in regional emission estimates of 447 ± 133 kt $CH_4$/year (Gaussian) and 347 ± 103 kt $CH_4$/year (mass balance), providing a good comparison with both the E-PRTR inventory and previous results from Fiehn et al. (2020) and Kostinek et al. (2021). However, the differences are not significant when the relatively large uncertainties are considered. Upscaled regional emissions for $CO_2$ amount to 0.2 - 0.3 Mt $CO_2$/year for both quantification approaches, and represent only ~ 1 % of 510 the reported inventory and regional $CO_2$ estimates from Fiehn et al. (2020), confirming that the coal mine ventilation shafts are not a minor contributor to the regional $CO_2$ emissions.

The uncertainty in the emissions quantified by UAV-based AirCore measurements is linked to the stability of the wind, as discussed in Andersen et al. (2021). The 10-12 minute snapshots are not instantaneously sampled, and an unstable wind 515 may cause the emission plume to meander across the plane. Although a single flight may not accurately represent the





ventilation shaft emissions, this study shows that with multiple flight quantifications for a single shaft a good estimate of the shaft's emission rate can be made. Short-term flights over the span of two weeks are used to estimate an annual average, where emission rates may vary week-to-week. The regional emission estimates assume that all shafts of a single coal mine emit an equal amount, which clearly is not true. A more accurate up-scaling model taking into account the individual emission size of different shafts would help improve this estimate.

The use of UAV-based active AirCore measurements in combination with the inverse Gaussian approach and the mass balance approach has been demonstrated to be able to quantify the emissions from individual ventilation shafts, which can then be used to estimate regional emissions of both $CH_4$ and $CO_2$. The uncertainty of the regional estimates can be reduced by increasing the number of quantified shafts. The UAV system is flexible and versatile, and opens up opportunities to quickly obtain regional estimates in regions that are otherwise hard to access. Be it the determination of a single emitting point source or a regional estimate, the UAV-based active AirCore system can be a valuable tool to help understand the $CH_4$ budget, and verify and constrain uncertainties of single strong $CH_4$ point source emitters or regions.





# 6 Supplement information

**6.1 Flight profiles**

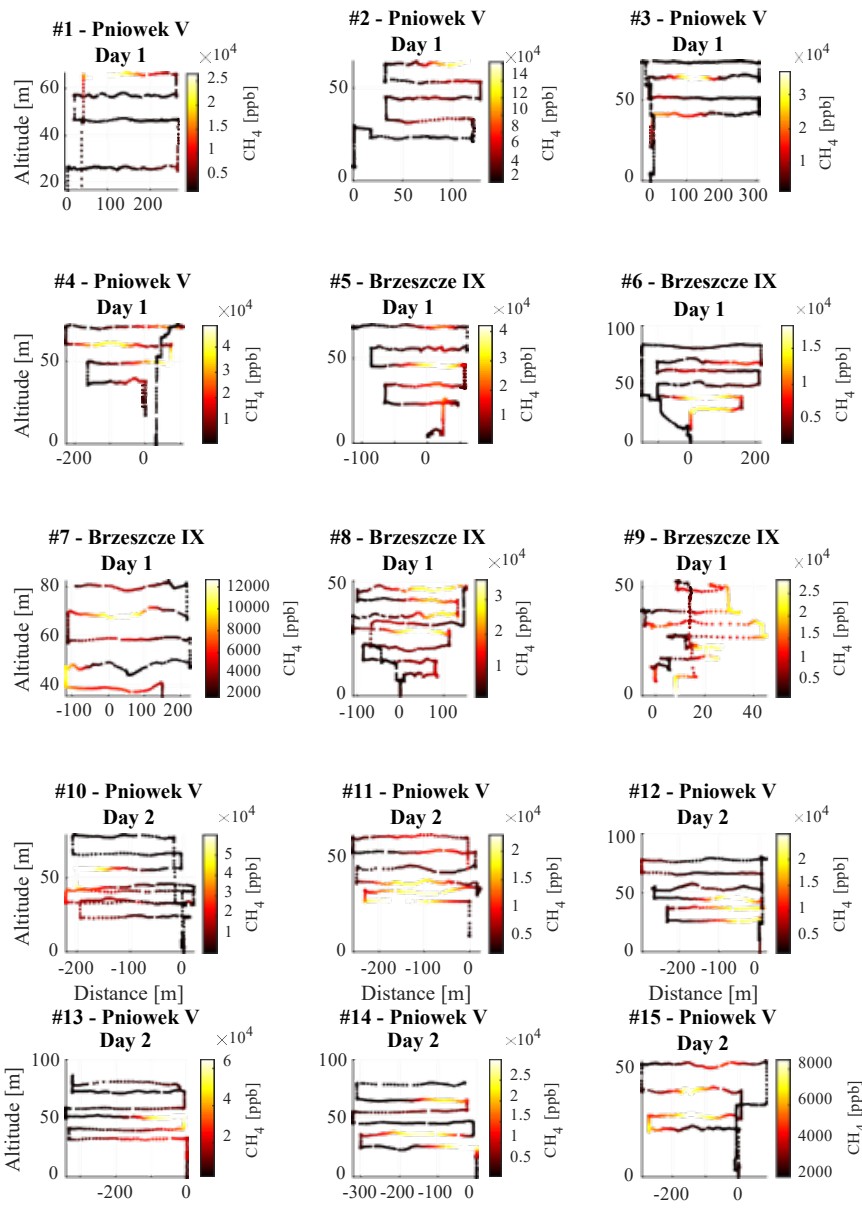

**Figure 12.** The measured flight profiles for flights #1 to #15.





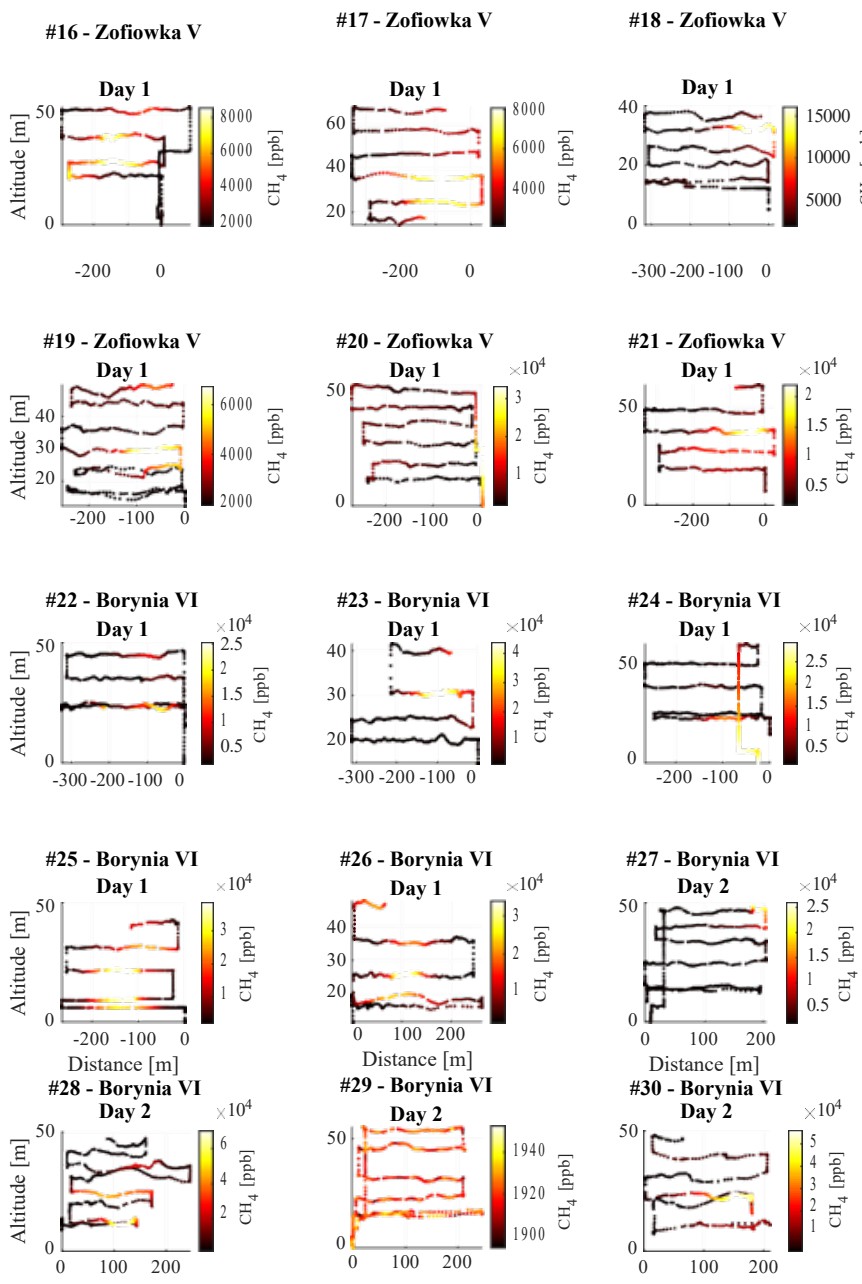

**Figure 13.** The measured flight profiles for flights #15 to #30.





**Figure 14.** measured flight profiles for flights #31 to #45.





**Figure 15.** measured flight profiles for flights #46 to #59.





*Data availability.* The raw data sets and flight logs, as well as wind data from the period May 18 – June 1 (2018), can be accessed at https://doi.org/10.5281/zenodo.5786532 (Andersen et al., 2021).

*Author contributions.* HC, TA, AR planned the campaign; TA, MdV, HC, MM performed the measurements; TA and HC analyzed the data; TA and HC wrote the manuscript draft; WP, JN, JS, MM, TR, AR, AF reviewed and edited the manuscript.

*Competing interests.* The authors declare that they have no conflict of interest.

**Acknowledgements.** This study is partially funded by the MEthane goes Mobile: MEasurement and MOdeling (MEMO[2]) project from the European Union's Horizon 2020 research and innovation programme under the Marie Sklodowska-Curie grant agreement No 722479. We would like to thank the CoMet project for the opportunity to participate in an exciting and stimulating campaign and collaborate with the participants of the campaign with tons of great discussions and good times.

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
