# Peer review of "Local to regional methane emissions from the Upper Silesia Coal Basin (USCB) quantified using UAV-based atmospheric measurements"

_Atmospheric Chemistry and Physics, 2021_

## Author Comment (AC1)

Thanks for giving us an opportunity to submit a revised version. We have improved our manuscript based on the constructive comments from the reviewers. Below you find one-to-one responses to all comments raised by the reviewers (our responses in blue). We are confident that we have addressed the concerns from the reviewers and hope that our manuscript will be accepted for publication.

**RC1**: 'Comment on acp-2021-1061', Anonymous Referee #1, 24 Jan 2022

**General Comments:**

In this paper, Andersen et al. use an AirCore UAV system to quantify $CH_4$ and $CO_2$ emissions from coal vent shafts in Poland using two different methods. These quantified results are compared with directly measured (in stack) emissions, both hourly and aggregated by vent/day. Andersen et al. then use various techniques to upscale the quantified emissions in order to obtain regional emission estimates.

Overall, this work has important implications in the field of methane quantification from coal vents. However, there are non-negligible gaps in the science presented here. Most notably, the authors do not adequately explain the effect of and error introduced by flights when the maximum detected concentrations are on the edge of the "curtain", indicating that the peak plume concentrations may not have been sampled. Furthermore, the manuscript lacks an important figure which directly compares each flight-based quantification with direct (in stack) measurements. The manuscript can also be improved with a clearer, more coherent argument for the potential impact the manuscript will have on the state of this science. With these changes, this manuscript will be a valuable addition to the literature surrounding methane quantification, AirCore viability, and upscaling procedures. I was glad to get to read this manuscript and provide (hopefully) helpful feedback.

AR: We are pleased to see the encouraging comment. As the concerns are also raised in the specific comments, we respond to the concerns below.

**Specific Comments:**

I believe there is a lack of explanation in the methodology section regarding the quantification procedures.

AR: Yes, we have added a more detailed description of the quantification procedures. Please refer to our responses to the specific comments below.

For the inverse gaussian approach, what point(s) are plugged into the equation? Are multiple points used and compared/averaged?

AR: All the points collected by a single flight like the one shown in fig 2(a) are plugged into the equation (1). We did not average the flight data points but rather smoothed the simulated plumes according to the spatial resolution of the AirCore measurements. See also our response to the question below about the effect of AirCore smoothing on the quantification.

Is the maximum concentration used and assumed to be the center of the plume? How are the dispersion parameters determined (what method), and how do they affect results?

AR: No, we do not use or assume the maximum concentration to be the center of the plume. The dispersion parameters in the horizontal and the vertical direction ($\sigma_y$ and $\sigma_z$), together with the emission rate ($Q$) and the coordinates of the center of the plume in the curtain (height $H$ and distance D) are five unknown parameters that are optimized using the following method.

We have added the following paragraph in the revised version (the last paragraph in section 2.4):

"The AirCore flight data ($Y$) presented in fig 2(a) is compared with the plume simulations of the Gaussian dispersion model. A best fit for eq. 1 to the data can be found for these five parameters by minimizing the cost function $J(Q, \sigma_y, \sigma_z, H, D) = (C(Q, \sigma_y, \sigma_z, H, D) - Y)^2$ using a standard square error (SSE) approach. The five parameters include the dispersion parameters in the horizontal and the vertical direction ($\sigma_y$ and $\sigma_z$), the emission rate ($Q$) and the coordinates of the center of the plume in the curtain (height $H$ and distance D). A group of random starting points for the five parameters between their lower and upper boundaries are set for the optimizer each time, and the optimization is run 1000 times to ensure that not only a local minimum is found (Andersen et al., 2021). In this way, we obtain a series of optimized values for each of the four parameters as the final results, and the five unknown parameters are optimized simultaneously."

In fact, Andersen et al., 2021 already used the exact five parameters in the optimization scripts. However, the equation in Andersen et al., 2021 was written with four parameters without the distance parameter, D, which was a mistake that we only find out now.

Are concentration peaks dampened by the AirCore method due to mixing in the sampling tube before analysis, and how does this effect quantification?

AR: Indeed, the AirCore concentration peaks are dampened due to molecular and Taylor diffusions in the sampling tube, but mostly due to mixing of air samples in the cavity of the analyzer. Deconvolving the measured signal to obtain the unaffected concentration peaks is possible, as is done in Andersen et al., 2021. However, we have found that the

moving averages of the original data using an averaging kernel of 33–34 s can well match the convoluted signal. Therefore, the simulated data from the Gaussian model is smoothed with such an averaging kernel before comparing with the AirCore observations. This was thus performed for all flights during the data processing.

We have added the following paragraph to the revised version (the last paragraph in section 2.2):

"AirCore concentration peaks are dampened due to molecular and Taylor diffusions in the sampling tube, but mostly due to mixing of air samples in the cavity of the analyzer (Andersen et al., 2018). Deconvolving the measured signal to obtain the unaffected concentration peaks is possible, as is done in Andersen et al., 2021. However, we have found that the moving averages of the original data using an averaging kernel of 33–34 s can well match the convoluted signal. Therefore, the simulated data from the Gaussian model is smoothed with such an averaging kernel before comparing with the AirCore observations. This was thus used for all flights during the data processing."

A critical issue is how you address those flights where the maximum concentration is at the edge of the curtain. How are these flights interpreted? It is hinted at in section 3.2, but I'm confused as to how you are calculating either the IG or MB if the majority of the plume is outside the curtain. This may be clarified by some of the questions in the above paragraph. It would also be nice to see some type of error analysis for each quantification method; that is, how do things like wind variability, peak dampening, dispersion parameters, etc. introduce error and how is this error quantified.

AR: We have treated all flights that fulfill the sampling criteria presented in Andersen et al., 2021 in the same way. The criterion "the wind direction deviates from the vertical direction of the curtain is less than 15°" excludes flights where the maximum is outside the curtain and thus limits the number of flights where the maximum concentration is at the edge of the curtain. Furthermore, for some flights including those where the maximum concentration is at the edge of the curtain, the IG method fails because the optimization scheme won't find a solution. For the MB approach, biased results will be obtained because the missing information cannot be recovered, which means that our estimated emission rate will be underestimated.

We have added the following paragraph to the revised version:

"A detailed description of the uncertainty analysis for both the IG and the MB methods has been presented in Andersen et al. (2021). Here, we only give a brief description. The uncertainty of the IG method is calculated as the standard deviation of a series of optimized emission rates generated by a large number of optimization runs (N = 1000).

The uncertainty of the MB method is mainly determined by the uncertainty and the variability of wind speed and wind direction measurements."

In the same vain, I think there is some issue with how error is represented in the aggregate data. For example, in the aggregation of quantified flux from Pniowek IV (Figure 5c) you claim an error of +-0.2kt/y due to the standard deviation of averaged points. However, in the individual day data for this vent (Figure 6c), the inherent error in each measurement is on the order of 3kt/y. A more robust error propagation analysis would make the aggregate numbers more defensible.

AR: In figure 5, we use the variability of multiple flights to indicate the uncertainty. Indeed, in figure 6, the error in each measurement is larger, because of systematic uncertainties that cannot be reduced statistically. We have discussed with a statistitian and got confirmed that the two types of errors cannot be simply merged as they represent different samples. We have therefore sticked to the use of the variabilitiy of multiple flights to indicate the uncertainty in figure 5.

A plot I'd really like to see is the hourly emissions compared with the flight quantified emissions (basically combining Figures 6 and 7). There is a bit of a roundabout comparison in the "hourly inventory" vs UAV quantified analysis (Figure 8b), but the critical representation is missing. The direct comparison is a key figure as it validates your UAV quantification approaches with real, empirical vent emissions data. As you state, emissions vary both intra and interday, so comparing UAV measurements at specific times with the directly quantified vent emissions instead of relying on aggregate data (like that presented in Figure 9) is an important distinction.

AR: We fully agree, and have added a direct comparison of the hourly emissions and the flight quantified emissions, and updated Figure. 6, see below.

We have updated the figure caption to "The quantified CH4 emission for each flight divided into different ventilation shaft and separated by individual flight days, with the hourly inventory. The emissions are also color differentiated by IG approach (red) or MB approach (blue). The number of quantifications on each day from the two methods is indicated in the parenthesis."

We have also updated the first sentence in section 3.2, "…, along with the hourly inventory presented the next section."

[Figure]

In section 2.1, you describe the del13CH4 data collection, but I am confused as how this is conducted. Are you capturing the outlet air of the Picarro upon measuring CH4/CO2 from the AirCore in a bag then analyzing? Some clarification would be helpful.

AR: Indeed, the air samples were collected at the outlet of the Picarro, downstream of the pump, and were stored in Tedlar bags for further analysis of isotopic signatures of $\delta^{13}$C-CH$_4$ and $\delta^2$H-CH$_4$ at a later time in the laboratory using a continuous flow isotope ratio mass spectrometer system.

We have modified and clarified the sample collection procedure in the main text in section 2.2.

Original:

"Directly after the CRDS analysis, the AirCore samples were collected in Tedlar bags for further analysis of isotopic signatures of $\delta^{13}$C-CH$_4$ and $\delta^2$H-CH$_4$. The isotopic composition was determined by analyzing the samples stored in the Tedlar bags using a continuous flow isotope ratio mass spectrometer system. More details about the analytical system and the calibration are provided in Brass and Röckmann, 2011; Röckmann et al, 2016; Menoud et al., 2021."

Revised:

"The AirCore samples were collected at the outlet of the Picarro, downstream of the pump, and were stored in Tedlar bags for further analysis of isotopic signatures of $\delta^{13}$C-CH$_4$ and $\delta^2$H-CH$_4$ at a later time in the laboratory using a continuous flow isotope ratio mass spectrometer system. More details about the analytical system and the calibration are provided in Brass and Röckmann, 2011; Röckmann et al, 2016; Menoud et al., 2021."

In my opinion, the argument of "weekends/holidays" does not add any value. If anything, it is confusing, as you postulate a reduction of emissions on these days then go on to show otherwise.

AR: We agree and have removed the "weekends/holidays" paragraphs in section 3.2 and 3.3.

Finally, I think there could be some added discussion about lessons learned and recommendations for future use of AirCore technology to quantify vent emissions. Specifically, expanding beyond *why* the hourly emissions data and individual flight quantifications may not align well and describing *how* the methods may be improved would be helpful. Similarly, some discussion of the recommendations for best practices to achieve a certain level of accuracy for quantifying vent/regional emissions using AirCore flights would be helpful; such as, how many flights are needed over how many days...etc.

AR: We thank the reviewer for the suggestion and have changed the conclusions section to "conclusions and outlook", and expanded the final paragraph. The paragraphs below have been added to the revised version.

"The use of UAV-based active AirCore measurements in combination with the inverse Gaussian approach and the mass balance approach has been demonstrated to be able to quantify the emissions from individual ventilation shafts, which can then be used to estimate regional emissions of both CH4 and CO2. However, the uncertainty of individual flight quantifications may be large, due to variable wind conditions under complexed turbulent schemes. Also, the in situ plume sampled by the AirCore does not necessarily follow the assumed Gaussian dispersion, as the averaging time is not sufficiently long, i.e., less than 30 minutes, which inevitably increases the uncertainty of the estimates by the inverse Gaussian method. To this end, optimization schemes that do not rely on the simple assumption of a Gaussian dispersion may be valuable (Shi et al., 2022). On the other hand, the complexed dispersion of the plume can be simulated by 3D large eddy simulation (LES), which can provide guidance to the design of the sampling strategy and

help develop a suitable method to estimate the emission rates based on the in situ plume sampling (Ražnjević et al., 2022).

The uncertainty of the estimates of individual shafts can be reduced by increasing the number of the quantification flights, although it is challenging to determine the exact number of flights needed to achieve a target uncertainty. Analysis of a large number of controlled tracer release experiments may provide an opportunity to directly address this issue, as has been performed for UAV measurements as well as many other different measurement platforms (Feitz et al., 2018; Bell et al., 2020; Morales et al., 2022).

Also, the uncertainty of the regional estimates can be reduced by increasing the number of quantified shafts. The limited number of quantified shafts makes our upscaling to the regional emission vulnerable.  Nevertheless, the UAV system is flexible and versatile, and opens up opportunities to quickly obtain regional estimates in regions that are otherwise hard to access. The UAV-based active AirCore system, thus, has shown to be a valuable tool to estimate $CH_4$ emissions on local to regional scales."

**Technical Corrections:**

16: Insert (CH4) after methane.

Done

23: Delete "have"

Done

28: Delete "though"

Done

28-29: Rephrase "As an alternative…" sentence. Make sure verb tenses match and phrasing is clear.

Done

34: Is methane the second "most abundant" or just second most important in terms of climate forcing?

Changed to "most important". Methane is actually also the second most abundant anthropogenic greenhouse gas after CO2, while H2O is not anthropogenic.

49: Citation for coal being 12% of methane emissions?

Added a reference "Saunois et al., 2020"

52: Change "part of" to "some"

Done

54: Change "releases" to "is released"

Done

56: Insert comma between "mines" and "the"

Done

58: Citation for data loggers lacking accuracy and temporal resolution? It seems that your data shows otherwise… high resolution and temporally resolved fluxes from vents.

We have changed it to "lacking accuracy and continuity (Swolkień, 2020)".

64: Sources for other studies using UAVs for methane monitoring?

We've had references using UAVs for methane (GHG) monitoring in the sentences following this one.

71: Perhaps add a line describing the Merlin mission and how CoMet ties in?

We have changed the sentence to "The CoMet aims at preparing the validation activities for the upcoming German-French Climate satellite mission MERLIN," and added one more reference, Fix et al., 2018.

78: Change "strong ties to hard coal mining" to " containing extensive hard coal mining" or similar.

Done

83: Period after PRTR

Done

83: Remove "the" after "quantify" and before "emitted"

Done

Paragraphs 70-100: Ensure consistent verb tense. Example: 71 – "goal of CoMet *is to* provide", 76 – "CoMet campaign *was to* quantify" etc.

Done

86/89: It goes from 59 flights to 34 quantifications – consider adding a line about filtering and what section you discuss this, otherwise it is confusing why these numbers don't match.

Yes, we have changed the sentence to "Here we present quantified emissions of 34 active AirCore flights that fulfill the flight selection criteria (Andersen et al., 2021)…"

89: The quantified emissions are of the shafts using the aircore, not quantified emissions of aircore flights.

We have changed the sentence to "Here we present quantified emissions of shafts using 34 active AirCore flights…"

95-100: Consider removing the "Section 2 presents …, section 3 contains…" and instead replacing with a strong statement about what your results convey and why they are important.

Yes, we have removed the "Section 2 presents …, section 3 contains… A conclusion is given in Sect. 4." We added "We show that a strong correlation ($R^2$ = 0.7 – 0.9) was found between the quantified and hourly inventory data-based shaft-averaged $CH_4$ emissions. Based on the correlation, we estimated regional $CH_4$ emissions by upscaling shaft-averaged $CH_4$ emissions. "

123-124: The names of the vent shafts have not yet been introduced and I did not know what these names meant. Consider revising to introduce the region and vent shafts before this section (maybe move section 2.3 to beginning of methodology).

We have moved section 2.3 to the beginning of the methodology section.

126: "First few": specify how many.

We have changed it to "first four".

126: What meteorological parameters were collected?

We changed the sentence to "…meteorological parameters (ambient temperature, pressure, relative humidity, wind speed, and wind direction) were measured using a radiosonde (Sparv Embedded AB, Sweden, model S1H2-R) identical to the one used in Andersen et al. (2021). "

129: Add "Meteorology for flights #5 through …"

Done

Section 2.2: Add details about the height of the meteorological sensors.

We have added the detail "at about 1.5 m above ground."

135-137: "The CSAT3 has an operating temperature … small changes in wind direction" is unnecessary.

We have removed the sentences.

144: Give some highlights about what the sampling criteria were to consider a "good flight"

We have added the details to the  "the mean wind speed during the flight is larger than 2 m/s and that the flights are performed perpendicular to the wind direction (within 15°)."

144: The intro said 34 flights were used for quantification, this line says 36 fulfilled the criteria – why the discrepancy?

Changed to 34

146: Add "technique" between "this" and "effectively"

Done

153-154: Add the altitude range for the flight to go with duration and downwind distances.

Add "altitudes up to 100 m above ground" after "The flight duration varied between 8 and 12 minutes".

Figure 1: Is there any reason for the different colors for each vent shaft? If so label.

Just to distinguish the different mines.

179: How do you account for plume rise? In the gaussian equation, I believe his typically the "effective stack height" which accounts for advective or buoyancy rise effects of the plume.

The inverse Gaussian model is based on the assumption of steady state and can't address plume rise. We have added the plume center height as a variable along with emission rate and dispersion parameters in horizontal and vertical direction to the optimizer that is described above. See also our responses to the general comment above.

Section 2.4: How is the local/regional background accounted for?

The minimum concentration of the entire flights was used as background, which was subtracted from the measured concentrations before calculation of the emissions for both the MB and the IG approach. The minimum concentration is not the same as a typical choice of e.g., 10 percentile; however, the two values are close and do not add significant errors as the $CH_4$ enhancements are very large.

We have added the following sentence: "The minimum concentration of the entire flights was used as background, which was subtracted from the measured concentrations before calculation of the emissions for both the MB and the IG approach."

196: Add "estimate" after "annual emission". Also, a source citing the E-PRTR inventory would be helpful.

We have added "estimate" after "annual emission", and the reference to the E-PRTR inventory Gałkowski et al., 2021 was already there.

202: Add comma after "active shafts"

Done

210-211: How do you account for the fact that the operating range of the sensors is <100% RH, but the conditions are often over 100%?

We have removed the sentence "The conditions are often rough and the relative humidity is high, and the readings of relative humidity could exceed 100% when the filter is wet."

215: Should "concentrations" be changed to "fluxes"?

No, here it should be "concentrations" instead of "fluxes", and the emissions rate (flux) is given in the equation below.

243-244: The sentence "All the isotopic...." Does not make sense.

The sentence "All the isotopic signatures found from the UAV active AirCore flights" has been removed.

Figure 6: I'm confused by the color differences – did different flights use different approaches (MB or IG)? I thought each flight was analyzed in both ways? If not be more clear in section 2 about this. Label what the error bars represent. Consider making the x axis on (b), (c), and (d) so that there isn't so much white space (restrict to sampling time period). Put in caption what the "N:7-5" means. Overall, I think there may be a better way to represent this data, consider reframing.

Both the IG and MB approaches have been applied to all flights that fulfilled the criteria. The missing quantifications from the IG method for some flights are entirely due to failures of the optimization. We have updated Figure 6.

298-299: "The Borynia VI inventory 'may therefore not represent...'" I'd think it clearly does not, given the intra and inter day variability in your other data.

We agree. However, we do not have a better way of getting the inventory estimate for Borynia VI.

Figure 8: Many of the labels are obscured, overlapping, or otherwise can't be read.

The figure has been updated. See below.

[Figure]

356: "Best statistics" – do you just mean the most flights? If so say that, if not clarify what "best statistics" means.

We have changed "best statistics" to "the most flights".

363: Wording is confusing

We have changed the sentence to "This could be due to a lack of statistics in the number of quantifications or the possible biases of the measured hourly inventory."

366: Again, is "lowest statistic" just fewest flights?

We have changed "lowest statistics" to "the fewest flights".

370: "All over"? Confusing

We have changed it to "Thus, the measured distributions for Pniowek V, Pniowek IV, and Zofiowka IV overlap with the hourly inventory distributions"

421, 424, others: Replace "linear curve" with "line"

Done

421: Comma between "rate" and "calculated"

Done

450-456: Instead of "comparing" to estimates then talking about how the estimates don't include coal, perhaps introduce this idea earlier. In reading, it is confusing why the numbers are so different until I realized that the EPRTR estimate really doesn't represent coal emissions at all.

We have added the following two sentences before discussing the coal-related emissions. "According to Swolkien, 2020, there are collocated $CO_2$ emissions along with $CH_4$ emissions during the extraction of coal. However, $CO_2$ emissions from coal mining activities are not included in the E-PRTR inventory."

462: Add "method" after "upscaling"

Done

466/471: These lines contradict one another. "does not accurately represent emissions of the whole region" vs "a useful tool for regional emission estimates". Best to clarify.

Here we meant to say that a simple approach by grouping the measured five shafts to obtain the average does not accurately represent the emissions of the whole region. However, using the slope and intercept of the strong linear correlation may provide a useful tool for regional emission estimates. This has been indicated in the paragraph "especially using the third approach of deriving the quantified emissions from hourly inventory data and scaling this to a regional emission rate"

500: Delete "have" between "we" and "used"

Done

S.I.: The color scale makes it so that the peak (and most critical part) of each plume is invisible (white).

Thank you for pointing out the issue. We have updated the colormap in Figure 2 and Figure S1-S4.

---

## Author Comment (AC2)

Thanks for giving us an opportunity to submit a revised version. We have improved our manuscript based on the constructive comments from the reviewers. Below you find one-to-one responses to all comments raised by the reviewers (our responses in blue). We are confident that we have addressed the concerns from the reviewers and hope that our manuscript will be accepted for publication.

**RC2**: 'Comment on acp-2021-1061',

Review of "Local to regional methane emissions from the Upper Silesia Coal Basin (USCB) quantified using UAV-based atmospheric Measurements**"**

**General comments:**

The authors present a manuscript following a measurement campaign in the Upper Silesian Coal Basin in Poland, where they used an AirCore attached to a UAV in order to measure $CH_4$ and $CO_2$ concentrations from several individual coal mine shafts. They make the case that a regional estimate derived from shaft-specific measurements will be superior to those that assign a single number to each mine, which broadly takes an average number across all shafts at any mine. They find good agreement between their methane measurements and high-resolution hourly inventory data in some of the shafts, whereas their flights were not able to reproduce well the coarser inventory numbers based on yearly estimates. They also claim that their $CO_2$ measurements have found that coal mines may be an overlooked source in the region that is not insignificant.

Overall, the paper is well-written and their ideas are clearly presented. The $CH_4$ analysis, in particular, is laid out in a straightforward manner that is easy to understand. That said, I feel that there are some important elements that are missing and some important changes that need to be made to this manuscript before it will be ready for publication. In particular, I have strong concerns about the $CO_2$ analysis, as I do not feel that there is enough data presented to support the conclusion that they have found a missing source of ~1% from the regional inventory, rather than it possibly being an artefact of upscaling. This, combined with my concern about how there is no independent value to compare the $CO_2$ measurements against, and some other smaller concerns I detail later, means my inclination right now would be to recommend leaving the $CO_2$ analysis out of this manuscript entirely.

AR: We thank the reviewer to point out the concern about the $CO_2$ analysis. First of all, we would like to clarify that the message was not that we have found a missing source of ~1% from the regional inventory, but rather we found a potential way of estimating $CO_2$ emissions from coal mine ventilation shafts and found it to be ~1% of regional $CO_2$

emissions. The upscaling to the regional $CO_2$ emissions is of course with assumptions, and we've tried to clarify this in the revised version.

Based on the strong correlation between observed $CO_2$ and $CH_4$, we derived the average $CH_4/CO_2$ slope of 4.6 $ppmCH_4/ppmCO_2$, which is consistent with the values found in Andersen et al., 2021.  The derived slope and the quantified $CH_4$ emissions for individual shafts are used to calculate the estimated $CO_2$ emissions. On the shaft scale, the uncertainty of estimated $CO_2$ emissions is comparable to that of estimated $CH_4$ emissions. The upscaling of regional $CO_2$ emissions is tied to the upscaling of regional $CH_4$ emissions. We have rephrased the $CO_2$ part in the revised version to focus on the estimated regional $CO_2$ emissions and the uncertainty of the regional $CO_2$ emissions instead of a missing source of ~1% from the regional inventory.

We have removed the following sentence from the main text" Due to the omission of $CO_2$ emitted from underground coal mining in the E-PRTR inventory, we conclude that the $CO_2$ inventory is missing a source of roughly 1 %.".

I also may have found an issue with the stated method of how the hourly methane emission rate is calculated, for those numbers that they later compare their UAV measurements against and find relationships with, which I would like to see clarified.

AR: We have double checked the calculation of the hourly methane emission rate and found no issue with the calculation. Specific clarifications will be provided based on the individual comments below.

And I would also like to see further explanation/justification for the first of the three presented upscaling methods, which upscales the yearly inventory numbers using a relationship identified only with the hourly inventory.

AR: We have provided the requested explanation and we refer to our responses to the specific comment on Lines 407-412.

 Further, I would like to see an expanded discussion of the possible areas of uncertainty including: i) the dangers of upscaling with such a small population of data (which they already do acknowledge briefly), ii) the uncertainties and potential misquantifications inherent in their plume calculation methods (the inverse Gaussian and especially the kriging), iii) the possibility of difficulties introduced by sampling at different times of day and under different atmospheric mixing conditions, and iv) at least some discussion of how the background was defined when calculating the leak rates, along with other ideas the authors may think of themselves.

AR: Thank you for this list of discussion items. We have tried to address each, while staying within a reasonable length of the discussion by making use of our earlier publications where possible. We have added the following paragraph to the revised version:

"A detailed description of the uncertainty analysis for both the IG and the MB methods has been presented in Andersen et al. (2021). Here we only give a brief description. The uncertainty of the IG method is calculated as the standard deviation of a series of optimized emission rates generated by a large number of optimization runs (N = 1000). The uncertainty of the MB method is mainly determined by the uncertainty and the variability of wind speed and wind direction measurements."

We have added the following sentence to acknowledge the small population of data: "We acknowledge that potentially large biases may have been introduced to the upscaling as the number of quantified shafts (5) is small compared to the total number of shafts (59)."

The minimum concentration of the entire flights was used as background, which was subtracted from the measured concentrations before calculation of the emissions for both the MB and the IG approach. The minimum concentration is not the same as a typical choice of e.g., 10 percentile; however, the difference of the two values is relatively small compared to the large $CH_4$ enhancements, and thus causes negligible difference in the calculated $CH_4$ emissions.

We have added the following sentence: "The minimum concentration of the entire flights was used as background, which was subtracted from the measured concentrations before calculation of the emissions for both the MB and the IG approach. The minimum concentration is not the same as a typical choice of e.g., 10 percentile (Vinković et al., 2022); however, the difference of the two values is relatively small compared to the large $CH_4$ enhancements, and thus causes negligible difference in the calculated $CH_4$ emissions."

I am curious, as well, as to whether the experimental set-up may mean that the AirCore samples are taken downstream of the rotors of the UAV, and whether that may introduce some dilution into their measurements (which may also help to explain why the measured values tend to be lower than the hourly inventory numbers).

AR: We appreciate this curiosity, and have been very careful ourselves in the design of this sampling system. The inlet of the AirCore system was positioned to the side of the carbon fiber box that is beneath the propellers. Therefore, the air sampled into the AirCore is effectively from above the propellers, within less than 0.5 m above the propellers (Lampert et al., 2020). As the UAV is most of time moving forward at a steady speed of 1-2 m/s, the collected air samples will not be disturbed . The change of the

effective sampling altitude for all transects on the order of 0.5 m has no significant impact on the quantifications.

We've added the following paragraph to the methodology section 2.1 UAV-based Active AirCore system in the revised version:

"The inlet of the AirCore system was positioned to the side of the carbon fiber box that is beneath the propellers (Andersen et al., 2021). Therefore, the air sampled into the AirCore is effectively from above the propellers, within less than 0.5 m above the propellers (Lampert et al., 2020). As the UAV is most of time moving forward at a steady speed of 1-2 m/s, the collected air samples will not be disturbed."

There are additionally smaller things that should be quicker to fix, but would also be essential, including double-checking the unit scale-factors on each figure that shows $CH_4$ mixing ratios (which often seem too high by a factor of 10) and the units on the $CH_4/CO_2$ ratios. I believe that starting with these changes will make a substantial impact on the quality of the manuscript, and that by the time it is ready for publication, it will be a valuable manuscript to the broader community.

AR: We understand that these high numbers confused the reviewer, but the $CH_4$ mole fractions were indeed that high, on the order of 100 ppm. We have double checked and can confirm the units presented in the manuscript.

**Specific comments:**

Lines 47-49: Is there a citation for the numbers in either of these 2 sentences? It's a key statement towards the motivation of the study—even referenced in the abstract—so I think it's important to show where those numbers come from.

AR: Thank you, we have added a reference "Saunois et al., 2020" for the numbers.

Lines 109-110: I am wondering how the AirCore was exactly "attached" to the UAV. I see that the AirCore is coiled up, and there is a reference a couple of lines up to "carbon fibre box housing". Is the AirCore contained within that housing, or is that just the housing of the electronics for the UAV? It would help if there was a picture showing the set-up. Particularly I am wondering how it was ensured that the AirCore was measuring from air that was undisturbed by the rotors of the UAV. It looks like this UAV has 4 vertical rotors, and if the AirCore is taking air from underneath (or otherwise "behind" the rotors), then there may be a risk that the rotors are mixing the air (potentially pulling in more dilute air from the background) just before measurement, and therefore affecting the measured mixing ratios. If so, I would be interested in knowing how much effect this may have on the ultimate measurements. And along those same lines, I would

wonder about what the effect on sampling rate is when the UAV is moving, considering the primary driver of intake is the ambient pressure. (Is the AirCore exposed in a way that it would sample more when the intake is pointed towards the direction of movement, because of the higher pressure, and vice versa? If so, how might that affect the results?)

AR: The AirCore is contained within the carbon fiber box housing beneath the propellers (Andersen et al., 2018). However, the inlet of the AirCore is positioned to the side of the box. The air samples are pulled into the AirCore through a pump at the outlet of the AirCore. The pumping flow rate is ~21.5 sccm. Besides this, the change of altitude affects the inlet pressure and thus the sampling. The two effects have been taken into account in the AirCore retrievals (Andersen et al., 2018). The possible ram pressure (=$0.5\rho v^2$) is on the order of 1 Pa with a flight speed of 1-2 m/s through static air, which is similar to the pressure change caused by a vertical displacement of 10 cm, and can be neglected during the AirCore retrieval.

We have added the following paragraph to the manuscript:

"The inlet of the AirCore system was positioned to the side of the carbon fiber box that is beneath the propellers (Andersen et al., 2021). Therefore, the air sampled into the AirCore is effectively from above the propellers, within less than 0.5 m above the propellers (Lampert et al., 2020). As the UAV is most of time moving forward at a steady speed of 1-2 m/s, the collected air samples will be fresh."

Figure 2: The units on these colorscales seem at least a factor of 10 too high. Were the authors really detecting plumes of 150 ppm of methane?

AR: The $CH_4$ mole fractions were indeed that high, on the order of 100 ppm of methane.

Line 210: This methane sensor gives output as a percentage concentration? Am I understanding that correctly?

AR: Yes, the methane sensor indeed gives output as a percentage concentration.

Line 213: I might be misunderstanding this sentence. It says "about 5% of the vented air to the atmosphere is from air inflow via the ventilation shaft closure". I understand that to mean that there is some quantity of vented air in this region, and that 5% of that total ventilated air comes from the shaft closure here. That does not sound like the same thing as saying 5% of the total gas flowing through this shaft gets vented. In order for it to contribute 5% of the total vented gas, we would need to know what the total vented amount is, then we take 5% of that number and use that to see how much of the gas

flowing through the shaft would have to be venting. So, have I misunderstood the statement here? If not, then the "95% of the flow-rate" scaling factor would not work.

AR: Sorry that our statement was confusing and apparently misunderstood. The measured air flow contains not only the vented air from the underground coal mines, but also some additions from the ambient from the ventilation shaft closure that accounts for about 5%.

We have changed the sentence to

"According to the statements of ventilation engineers, the measured air flow includes about 5% ambient air from the ventilation shaft closure, …"

Lines 230-235: This is a lot of words to describe the math, and I think I got a bit lost. Would it be possible to include the simple formulas for these 3 upscaling techniques?

AR: Thanks for the good suggestion. We have described the three upscaling methods using equations as below:
$$Q_{M1} = Q_{E-PRTR-regional} \times k_1,$$
$$Q_{M2} = \bar{Q}_{UAV-shaft} \times n,$$
$$Q_{M3} = \left(\bar{Q}_{hourly-shaft} \times k_2 + b\right) \times n,$$
where $Q_{E-PRTR-regional}$ is the annual E-PRTR emission rate, $\bar{Q}_{UAV-shaft}$ is the mean quantified shaft emission rate, $\bar{Q}_{hourly-shaft}$ is the mean hourly inventory emission rate, $k_2$ and $b$ are the slope and the intercept of the linear fit of shaft-averaged emissions between our UAV quantified and high frequency (hourly) reported emissions, while $k_1$ is the slope of the linear fit that is forced through zero, and n is the number of active ventilation shafts in the region.
We have added the equations to section 2.6 in the revised version.

Line 282: I'm not sure that I'm convinced that there is a potential difference between weekend/holiday and weekdays, given the mass balance numbers. The inverse Gaussian numbers seem more like they could suggest that, but is there a reason to trust these more than the mass balance numbers? Feels like one shouldn't hint at a conclusion either way. (I assume that the phrasing "this may indicate…" is maybe an attempt to stay neutral, but it still reads to me like it's leaning towards the conclusion that there is a relationship.)

AR: As mentioned in the reply to reviewer #1, we've removed all texts regarding the weekend/holiday and weekdays.

Figure 6: Maybe this shouldn't be explained in the caption, exactly, but I'm not finding where in the text it explains why certain flights were deemed worthy of a mass-balance estimate but not of a Gaussian estimate?

AR: The reviewer raises a valid question here. The two methods have been applied to all flights that fulfilled the criteria. The missing quantifications from the IG method for some flights are entirely due to failures of the optimization. Please also refer to our responses to the comment by reviewer#1 "A critical issue is how you address those flights where the maximum concentration is at the edge of the curtain".

We've added the following sentences at the end of the first paragraph in section 3.2 Quantified $CH_4$ emissions.

"Note that both the IG and MB approaches have been applied to all flights that fulfilled the criteria. The missing quantifications from the IG method for some flights are entirely due to failures of the optimization."

And the following sentence to the caption of Figure 6.

"The number of successful quantifications on each day from the two methods is indicated in the parenthesis."

Line 297: Instead of assuming, is there anyone who could be contacted/referenced that would have more insight into why this period is missing from the inventory data?

AR: Our collaborators in Poland, also co-authors of the manuscript, have tried to obtain more details, but unfortunately it is not possible to obtain more info.

Line 305: So the inventory seems to contradict the hypothesis that there's a difference between weekend/holiday and weekday emissions. To me, though, this seems consistent with the lack of conclusions we could have drawn from the data, anyway?

AR: As mentioned above, we've removed all texts regarding the weekend/holiday and weekdays.

Figure 7: Looking at Pniowek V, for example because it has the longest timeseries, the inventory would lead me to expect higher measured values on the 19th, 21st, and 28th compared to the 31st and June 1st, but that's not exactly what was seen in Figure 6, which shows low values recorded on all of the flights of the 28th and potentially high values on June 1st. Do we have an explanation for this discrepancy? (I actually think it might have been nice to combine Figures 6 and 7, so that we see the overlay of the measurements against the reported inventory directly.)

AR: Thank you, we have updated Figure 6 as suggested by the reviewer, see below. From the updated figure, we can see that the inventory estimates and the UAV quantified emission rates are broadly consistent within large uncertainties.

[Figure]

Line 313: Wouldn't we expect that the correlation between individual flights and yearly reported emissions would be very low, though? Because day-to-day variability would be so high, in comparison?

AR: Yes, the correlation between individual flights and yearly reported emissions would be expected to be low, and is in fact low according to our analysis.

Table 2: Could we convert this to a bar chart, maybe? (One could mark the max/min values separately from the error bars, and include the N numbers at the tops or bottoms of the value bars.)

AR: Thank you for pointing out the duplication. The relevant information is already presented in figure 9. Therefore, we have moved table 2 to the supplemental info.

Figure 8: It's difficult to intuit where the 1:1 line would be with rectangular figures like this. Would it be possible to make these figures square with identical limits on the axes,

to really visualize the comparison? Maybe with a dashed 1:1 line, for reference? (I understand that this might necessitate dividing this up into 2 figures, in order to fit on the page.)

Figure 8: It also may be helpful to change the legends of each subplot to indicate that it's the best-fit line from the inverse Gaussian approach, specifically, as is noted in the caption

AR: Thank you for the helpful suggestion, we have indeed updated Figure 8, and added a dashed 1:1 line for comparison. See the figure below.

[Figure]

Line 331: What is the justification for forcing the linear fit through 0?

AR: This is to avoid of inferring an unrealistically large quantified estimate when the inventory estimate is approaching zero. We have provided two types of linear fits, with and without forcing the linear fit through zero. The linear fits for larger sources do not change as much as for near-zero sources.

Line 339: The hourly inventory is going to be used to scale up the UAV-measured concentrations?

AR: Indeed the reviewer is correct, the hourly inventory is used to scale up the UAV quantifications. Due to the relatively low correlation between the hourly inventory and the quantified emission from individual flight, we have used the correlation between the shaft-averaged hourly inventory and UAC-quantified emission rates to scale up.

Line 341: Of the linear fit from the multiple-days-averaged shaft-specific, inverse Gaussian case?

AR: We have used both the inverse Gaussian and the mass balance estimates for scaling up, and have changed the sentence to "We use the slopes and the intercepts found in Figure 8c to scale up our quantified emissions."

Figure 9: Is this all the same info from Table 2, it seems?  If so, maybe we can just get rid of Table 2 and refer to this instead?

AR: Indeed it is from Table 2, which is moved to the supplement based on the reviewer's earlier recommendation.

Lines 358-9: Does this also imply that the sample size might not be enough to accurately quantify the other sites?

AR: A good question indeed, but we can't really provide a specific number for the sample size that is enough to accurately quantify the emissions. More flights will certainly provide better statistics. We've removed the sentence "This indicates that this statistical pool is sufficient to accurately quantify comparable emissions".

Line 362: It doesn't look to me like there is overlap at Pniowek IV in the mass balance approach...?

AR: True, we've removed "the mass balance approach" from the sentence.

Lines 374-5: I think here is where to mention the possible explanations for lower quantification in the air than what the hourly measurements within the shaft show, rather than lines 362-364, which was specifically talking about Pniowek IV

AR: Thanks for the suggestion. We have moved the possible explanations in lines 362-364 to lines 374-5 of the original version, and changed the sentence to "This could be due to a lack of statistics in the number of quantifications or the possible biases of the measured hourly inventory"

Line 382: One thing I don't think I understand is, if $CO_2$ has been measured as well as $CH_4$ from the AirCore, then why not just calculate the emission rate of $CO_2$ in the same way as was done with $CH_4$? Why introduce some linear dependence with methane and throw away the data that does not sufficiently have that linear dependence? Is the thinking that, if there are enhancements seen with $CH_4$, then it's presumed to originate from the shaft, but if there are enhancements in $CO_2$, they could also be from elsewhere nearby (are there other $CO_2$ sources nearby, like running engines?)? So this is done in order to ensure that one only looks at $CO_2$ that is believed to be from the shaft?

AR: We've used the linear correlation between enhanced $CH_4$ and $CO_2$ to calculate the $CO_2$ emissions instead of directly using the $CO_2$ data for two reasons: 1) the $CO_2$ signal is relatively small compared to its variabilities, which makes it difficult to find a robust background signal; 2) it is indeed as the reviewer mentioned that we aim to quantify the $CO_2$ emissions from the shaft only.

We have added the following sentence to the end of the paragraph in the revised version:

"We've used the linear correlation between enhanced $CH_4$ and $CO_2$ to calculate the $CO_2$ emissions instead of directly using the $CO_2$ data for two reasons: 1) the $CO_2$ signal is relatively small compared to its variabilities, which makes it difficult to find a robust background signal; 2) we aim to quantify the $CO_2$ emissions from the shaft only."

Line 383: The authors probably should specify which is the numerator and denominator in "slope", even if it seems obvious.

AR: Thank you, we added "($CH_4/CO_2$)" at the end of the sentence "the slope is the slope of the linear fit between $CO_2$ and $CH_4$ ($CH_4/CO_2$)".

Lines 385-6: Would it be possible to include these scatter plots in the supplemental info, as well? I'm curious to see what they look like.

AR: Yes, we've added the scatter plots in the supplemental info, 5.2 Scatter plots of $CH_4$ and $CO_2$.

Line 387: I'm assuming the units are supposed to be ppb/ppm and not ppm/ppm? Additionally, this caused me to look at the figures in the supplemental info, where the flight tracks are provided, and it looks like the scaling factor on many of the colorbars is listed as $10^4$, but it should be $10^3$, since background methane should only be around 2ppm, not 20ppm.

AR: It is indeed ppmCH$_4$/ppmCO$_2$. The CH$_4$ mole fractions were indeed that high, on the order of 100 ppm of methane. However, the background was around 2 ppm, instead of 20 ppm, which can be seen from the updated color scale in Figs. S1-4 with updated colormap.

Line 392: Can the authors explain the NaNs again here? If there's not enough data to include an upper and lower bound, maybe it's better just to state that than to present it as a NaN value.

AR: The error bar is indicated as NaN when only one estimate is available. We have added it to the caption of figure 10.

Lines 407-412: I don't think I'm following the logic here. Figure 8 showed that there was no clear linear relationship between the measurements and the E-PRTR inventory, but that a relationship may instead be found when comparing against the hourly inventory. Then, here, the linear relationship that was found between the hourly inventory and the measurements is used to scale the E-PRTR inventory? What's the rationale for that?

AR: We acknowledge that this part is easily confusing for a reader, and we tried to explain it better in the revised text.

The E-PRTR inventory for each coal mine was provided, but the E-PRTR inventory for individual shafts were obtained by dividing the inventory for individual coal mines by the number of active shafts, which may have introduced large errors and may explain the very low correlation between the shaft-averaged E-PRTR inventory and UAV-quantified emissions. Here we scale the E-PRTR annual inventory for all shafts, assuming that the correlation between the shaft-averaged hourly inventory and UAV-quantified emissions are representative for the whole basin.

We have added the following sentences to section 3.5 Upscaling to regional estimates in the revised version:

"Here we assume that the correlation between the shaft-averaged hourly inventory and UAV-quantified emissions are representative for the whole basin and that the very low correlation between the shaft-averaged E-PRTR inventory and UAV-quantified emissions is mainly due to large errors introduced to the E-PRTR inventory for individual shafts by dividing the inventory for individual coal mines by the number of active shafts."

Lines 414-419: Might want to include an acknowledgement that the number of sampled shafts is small compared to the total number of shafts in the region (and among those sampled, those that have a large number of samples is even lower), so they may not be representative of the region as a whole.

AR: We fully agree, and have added the following sentence to the revised version:

"We acknowledge that potentially large biases may have been introduced to the upscaling as the number of quantified shafts (5) is small compared to the total number of shafts (59)."

Lines 421-2: My comment from the last paragraph should apply here, too. Though I think this is a much more sound approach than the first approach (which I would be tempted to toss out altogether without a clearer justification for why the hourly linear relationship would be directly applicable to the E-PRTR estimates).

AR: As explained above, we assume that the very low correlation between the shaft-averaged E-PRTR inventory and UAV-quantified emissions is mainly due to large errors introduced to the E-PRTR inventory for individual shafts by dividing the inventory for individual coal mines by the number of active shafts.

Line 441: When saying that they aren't statistically different when factoring in the uncertainties, should probably also acknowledge that the uncertainty bars are around 30%, which can be quite large.

AR: We fully agree and have added the following sentence to the revised version: "..., although the uncertainties are as large as 26-45%."

Lines 448-460: This illustrates the danger of upscaling to a region from just a few measurements. The authors note that coal mining activities are not a major source of $CO_2$ in the region, and that their measurements are also very low. The flight paths for the $CO_2$ enhancements are not included, so it's not apparent how clear or strong the $CO_2$ plumes really are compared to the background. Although Figure 10 shows that, though many of the quantifications do not have error bars, the ones that do are often quite large (e.g. Pniowek IV and Zofiowka IV). And since the E-PRTR inventory does not include coal mines in their inventory, there appears to be no way to independently check whether these values correspond to what would be expected or not.

AR: Based on our response above, we believe that the correlation found between observed $CO_2$ and $CH_4$ enhancements are strong and it is convincing to obtain the $CO_2$ emissions based on the estimate of $CH_4$ emissions. Also, because the $CH_4$ enhancements are very large, and the $CO_2$ enhancements are relatively small compared to its

variabilities, the coal mine related $CO_2$ emissions can only be obtained through the linear relationship.

Lines 475-476: I do not think that one can conclude that the $CO_2$ inventory is missing a source of about 1%. Without having more information presented about the nature of the $CO_2$ plumes that were quantified, it seems within the realm of possibility that contemporaneous $CO_2$ data recorded with the $CH_4$ data displayed some stochastic variations (especially if the atmosphere is not well-mixed) that could mistakenly be quantified as small plumes with the inverse Gaussian or kriging techniques, especially if the corresponding background values are not well defined. Then, by scaling up those small numbers to the size of the region, they become an apparently large number (~1%). But this feels to me more like a potential artefact of the upscaling than a real missing piece of the inventory. Would we otherwise have any reason to expect large amounts of $CO_2$ to come out of coal mines? (If so, this is something that I guess should also be addressed in the introduction?) Overall, it is starting to feel like it may be best to leave out the $CO_2$ analysis altogether.

AR: Besides what we have responded above, there is indeed a reason to expect large amounts of $CO_2$ from coal mines. According to Swolkien, 2020, $CO_2$ emissions accompany $CH_4$ emissions during the extraction of coal.

We have added the following sentence to section 3. 5 Upscaling to regional estimates.

"According to Swolkien, 2020, there are collocated $CO_2$ emissions along with CH4 emissions during the extraction of coal."

And have added the scatter plots of $CO_2$ and $CH_4$ enhancements from the flight measurements to the supplemental info.

Line 496: I thought it was only this large for 25 of the 36 flights? And again I think these units are incorrect.

AR: Thank you for spotting this, as there was indeed a mistake here and it should be 25 out of 34 flights. We have corrected it in the revised version. However, the unit is correct.

Lines 509-511: I really disagree with this conclusion without some compelling evidence that it's not just an artefact of the upscaling.

AR: We think the disagreement partly stems from the phrasing, which we discuss also in response to reviewer #1. Our intention was to present a way of estimating $CO_2$ emissions from coal mine ventilation shafts and put them into context, which is why we mention it to be ~1% of regional $CO_2$ emissions by other methods.

Line 516: Maybe the authors should point out that their data indicated that at least 5—and probably more—good flights were needed for a decent quantification of a single shaft.

AR: Thank you, a good point to add. We have added the following sentences to the revised version:
"The uncertainty of the estimate of individual shaft can be reduced by increasing the number of the quantification flights, although it is challenging to determine the exact number of needed flights to achieve certain uncertainty. Analysis of a large number of controlled tracer release experiments may provide an opportunity to directly address this issue, as has been performed for UAV measurements as well as many other different measurement platforms (Feitz et al., 2018; Bell et al., 2020; Morales et al., 2022). "
Lines 513-520: All of this (good) assessment of uncertainties should have, I think, belonged in the discussion section. It's fine to repeat it here, but it felt like it was lacking above. Additionally, included in the discussion of uncertainties should be a discussion of the inherent uncertainties involved in the techniques applied (especially with kriging, which can be a very uncertain way to quantify a plume!).

AR: We have moved the sentence "The uncertainty in the emissions quantified by UAV-based AirCore measurements is linked to the stability of the wind, as discussed in Andersen et al. (2021). The 10-12 minute snapshots are not instantaneously sampled, and an unstable wind may cause the emission plume to meander across the plane. " to section 3.2 Quantified $CH_4$ emissions. And the discussion on the uncertainties of the quantification has been added in section 2.4 Emission determination.

**Technical Corrections:**

Figure 3: One of the labels is cut off—the one attached to the red marker.

Done

Line 241: The isotope numbers should be in units of permil, not percent. It's correct in the figure, but not in the text. (May need to be corrected throughout the manuscript.)

Done

Line 300: "emitted emission" seems redundant

We have changed "emission" to "$CH_4$"

Line 328: "on an hourly basis"

Done

Line 332-3: "not significant"

Done

Line 338: "Our evaluations indicate"

Done

Line 370: "all overlap with"

Done

Line 374: replace "more than one flights" with "multiple flights"

Done

Line 381: remove "emission" from "emitted $CO_2$ emission"

Done

Figure 10: The caption describes these plots as "histograms". I do not believe that's the case.

Changed to bar plots

Line 403: "As many as"

Done

Line 448: "linear curves" should be "linear fits"

Done

Line 490: "show very low…"? agreement? correlation?

Added "correlation"

Lines 526-28: This last sentence feels like a long fragment instead of a complete sentence, and should probably be reworked

We have changed the sentence to "The UAV-based active AirCore system can be a valuable tool to estimate CH4 emissions on local to regional scales."

---

## Referee Report (RR1)

Round 2 Review of "*Local to regional methane emissions from the Upper Silesia Coal Basin (USCB) quantified using UAV-based atmospheric Measurements*"

**General comments:**

I am pleased to see that the authors made significant efforts to address each of the concerns that I raised in my first review. The manuscript now feels much clearer and more robust, overall.

The one issue where I do still have concerns is the $CO_2$ analysis. I do appreciate the authors' efforts to reframe this analysis away from the "1% missing inventory source" towards a simple reporting of the emissions estimate. However, I would like reassurance that the large uncertainty introduced by the linear fit for the ratio itself (2.9/4.6 = 63%) is being accounted for in the uncertainty values calculated and reported in the $CO_2$ analysis. I have the impression that this may have been overlooked, and will lead to much larger ranges on the estimates. These large uncertainties, then, would leave me finding it difficult to draw much meaning out of the analysis. Additionally, there appears to be little reference to the justification for the $CO_2$ analysis or to its conclusions in the abstract or introduction. (Though I suspect these may just be unintentional oversights.) Considering the potentially large uncertainties with such low emission values when compared to available regional estimates, I again raise the question of whether this $CO_2$ analysis is better off left out of the manuscript. That said, if the authors maintain that they want to keep this analysis in, I am much more comfortable with the conclusion at the end of the discussion section. There it states that, given how small and variable the $CO_2$ emissions are, these results support the idea that $CO_2$ emissions from coal mines are not significant, even from mines with rather significant $CH_4$ emissions. (Perhaps a comparison to a single automobile's expected annual emissions or something, for context, may help.) So if that is the takeaway that gets included in the abstract/intro, then I have no objection (after my question with the uncertainty calculations is addressed).

On the whole, I believe that this manuscript is in good shape to be a valuable contribution to the community, and will be ready for publication following some minor adjustments.

**Specific comments:**

Line 59: Here or somewhere else in the intro, it would be good to include the explanation for how and where $CO_2$ emissions come from during the coal extraction process, as motivation for their inclusion in this study. (Currently I see that one sentence had been added to Section 3.5, following a question I raised in the first round of reviews, but would be good to see that here, as well, and expanded on a bit.)

Line 97: Perhaps there should be some reference what comes out of the $CO_2$ analysis, too, if it is to be included in the manuscript?

Line 114: Somewhere in this paragraph, can the authors include a sentence or two about the time of day the measurements were taken and how well-mixed the atmosphere would be expected to be, including how that may affect the expected measurements?

Line 170: If the difference between the on-board met station and the off-site meteorological station was this high, was that incorporated in the uncertainty calculation for the plume calculations (in both approaches) for the flights that did not have the on-board met station (flights 5-33)?

Line 291: Can the authors include some examples for what causes optimization failures here?

Line 310: It may be good to mention here that attempts were made unsuccessfully to track down the real answer here, which is the reason for making this assumption.

Lines 350-357: Would be a good idea to specify in the caption descriptions which are the shaft-averaged measured emissions and which are the shaft-averaged inventory emission estimates. Otherwise using just "shaft-averaged emissions" for both axes gets confusing for figures b), c), and d).

Figure 8: Please specify with a legend which fit is being displayed here with the red line. Additionally, are the $R^2$ values really the same between c) and d)?

Line 378: I would like to see the authors include some additional possible explanations for the systematically low estimates here. Couldn't they also result from uncertainties in the quantification approaches or some poorly-accounted-for physical process? For example, the potentially large uncertainties in the wind speed and direction (especially for flights 5-33 that did not have truly local met data)? Or perhaps non-Gaussian plume behavior due to local turbulence?

Lines 388-9: Could there be some indication either in the text or in the appendix of which of the flights were kept for this analysis? Looking at the figures, I can take a pretty good guess about which ones were probably thrown out, but it would be better to state it plainly.

Lines 396-397: It's still not clear to me whether this is claiming that the $CO_2$ and $CH_4$ trends themselves are similar, which I would think would be baked into the use of the linear relationship for deriving the $CO_2$ values, and thus not a surprise.

Figure 10: Can the authors confirm that their uncertainty estimates incorporate the uncertainty associated with the CH4/CO2 slope? The 63% uncertainty on that linear fit slope is quite large, so I'm surprised to see some small error bars in the figure, e.g. in the Pniowek V chart.

Line 456: Can the authors articulate any other potential explanations for why the values presented here appear to be lower than in Fiehn et al.? (Maybe seasonality of emissions, for example?)

Line 481: Would specify for clarity "Estimated regional $CO_2$ emissions *from these coal mines*"

Line 517: I believe this should read that the "coal mine ventilations shafts **are** a minor contributor to the regional $CO_2$ emissions".

Lines 519-526: Maybe should include reference to seasonal changes, as well, and thoughts about the representativeness of 2 weeks out of the year to upscale an entire year's estimates.

Figures B1-B4: Can some indicator be included to highlight which flights were included vs. excluded from the different analyses? (Or if that's too much trouble, can this be stated in the text of the figure captions?)

Figures B5-B9: Similarly, can some indicator be included to highlight which flights were included vs. excluded in the analyses? Or can this be stated in the text? Additionally, please add the corresponding R2 values to each figure. And why are flights 31 and 56 missing?

**Technical comments:**

Line 23: Please define "IG" and "MB" before using the abbreviations

Line 67: Need a space between "Turnbull" and "et" in the citation

Line 104: There is a blue period and quotation mark here

Line 109-110: This sentence is repetitive of line 104

Line 112-113: This sentence about the "curtain" is also repeating what was just said a few lines above

Line 180: "downwind of the plume"

Line 283: "presented in the next section"

Line 355: There needs to be a space between "of" and "active"

Line 378: I'm not sure what happened to the spacing between the letters in the phrase "or the possible biases of"…?

Line 387: The $M_{CO_2}$ subscript needs to be fixed

Line 393: Unpaired quotation mark at the end of the line

Line 439: There is no need to include the "a" in "Fig. 11a"

Figure 11: The caption is counting the bars wrong after the second bar. It seems like the dark and light yellow were both counted as one bar, so all subsequent bars are described as one less than they really are (e.g. teal is actually bars 4 and 5, etc.)

Line 489: "an MB approach"

Line 504: Again, the ppm$_{CO2}$ subscripting looks weird

Line 524: Again, the spacing between the letters of the phrase "a single goal mine" does not match the rest of the text

Line 534: "complex" not "complexed"

Line 539: "of an individual shaft"

Figures B1-B4: Please either move the x-axis label to the bottom of the last row of figures or include it in all figures

---

## Author Response (AR3)

Round 2 Review of "*Local to regional methane emissions from the Upper Silesia Coal Basin (USCB) quantified using UAV-based atmospheric Measurements*"

**General comments:**

I am pleased to see that the authors made significant efforts to address each of the concerns that I raised in my first review. The manuscript now feels much clearer and more robust, overall.

AR: We thank for the reviewer's positive evaluation.

The one issue where I do still have concerns is the $CO_2$ analysis. I do appreciate the authors' efforts to reframe this analysis away from the "1% missing inventory source" towards a simple reporting of the emissions estimate. However, I would like reassurance that the large uncertainty introduced by the linear fit for the ratio itself (2.9/4.6 = 63%) is being accounted for in the uncertainty values calculated and reported in the $CO_2$ analysis. I have the impression that this may have been overlooked, and will lead to much larger ranges on the estimates. These large uncertainties, then, would leave me finding it difficult to draw much meaning out of the analysis.

AR: We understand that the reviewer has proposed an alternative method to estimate the total $CO_2$ emissions and their associated uncertainties. According to the reviewer's method, the estimated total $CO_2$ emissions equal the estimated total $CH_4$ emissions divided by the slope with the equation below:

$$Q_{CO_2} = \frac{Q_{CH_4} \cdot M_{CO_2}}{slope \cdot M_{CH_4}}$$

The corresponding relative uncertainty of the estimated total $CO_2$ will be

$$\frac{\sigma_{Q_{CO_2}}}{Q_{CO_2}} = \sqrt{\left(\frac{\sigma_{Q_{CH_4}}}{Q_{CH_4}}\right)^2 + \left(\frac{\sigma_{slope}}{slope}\right)^2}$$

We get $\frac{\sigma_{Q_{CO_2}}}{Q_{CO_2}} = \sqrt{\left(\frac{153.4}{324.5}\right)^2 + \left(\frac{2.9}{4.6}\right)^2}$ =79% for IG method and $\frac{\sigma_{Q_{CO_2}}}{Q_{CO_2}} = \sqrt{\left(\frac{188.8}{318.6}\right)^2 + \left(\frac{2.9}{4.6}\right)^2}$ =87%for MB method.

We have estimated the total $CO_2$ emissions based on the shaft-averaged $CO_2$ emissions, where the uncertainty was calculated based on the standard deviation of the $CO_2$ emissions from different shafts, in a similar way as we've done for $CH_4$. With this approach, we derived the mean estimates and their relative uncertainties:

2.2/4.2 = 52% for IG method and 2.3/3.8 = 61% for MB method.

The uncertainties derived from our method are smaller than what we understand the reviewer's method, however, we have not overlooked the large relative uncertainties of the slope.

The uncertainties of the estimated total $CO_2$ emissions in the main text were rounded to one digit, which artificially reduced the uncertainty. We have updated them to keep two digits: "which yields a regional emission estimate of $0.25 \pm 0.13$ Mt $CO_2$/year for the IG approach and $0.22 \pm 0.14$ Mt $CO_2$/year for the MB approach, respectively"

Additionally, there appears to be little reference to the justification for the $CO_2$ analysis or to its conclusions in the abstract or introduction. (Though I suspect these may just be unintentional oversights.) Considering the potentially large uncertainties with such low emission values when compared to available regional estimates, I again raise the question of whether this $CO_2$ analysis is better off left out of the manuscript. That said, if the authors maintain that they want to keep this analysis in, I am much more comfortable with the conclusion at the end of the discussion section. There it states that, given how small and variable the $CO_2$ emissions are, these results support the idea that $CO_2$ emissions from coal mines are not significant, even from mines with rather significant $CH_4$ emissions. (Perhaps a comparison to a single automobile's expected annual emissions or something, for context, may help.) So if that is the takeaway that gets included in the abstract/intro, then I have no objection (after my question with the uncertainty calculations is addressed).

AR: We thank the reviewer for the useful comment. Assuming that a typical automobile emits 7 liters or 18.9 kg $CO_2$ per 100 km and drives an average of 10,000 km per year, the total estimated $CO_2$ emissions of 0.25 Mt, 0.22 Mt would be equivalent to the emissions of ~130,000 and ~120,000 automobiles for IG and mass balance estimates, respectively.

In the introduction, we have added the following two sentences with references to justify our $CO_2$ analysis. "Meanwhile, the extraction of coal deposits is accompanied by emissions of other non-methane gases, including $CO_2$ (Swolkien, 2020). However, $CO_2$ emissions from coal mining are usually insignificant in terms of radiative forcing when compared with $CH_4$ emissions, and are therefore rarely quantified (Bonetti et al., 2019)."

In the results section 3.5, we have added the following sentence "…, and would be equivalent to the emissions of ~130,000 and ~120,000 automobiles (assuming 7 liters or 18.9 kg $CO_2$ per 100 km and an average of 10,000 km driving per year) for IG and mass balance estimates, respectively,…"

In the abstract, we have added the following sentence:" We have also estimated the total $CO_2$ emissions from coal mining ventilation shafts based on the observed ratio of $CH_4$/$CO_2$, and found that the estimated regional $CO_2$ emissions are not a major source of $CO_2$ in the USCB."

On the whole, I believe that this manuscript is in good shape to be a valuable contribution to the community, and will be ready for publication following some minor adjustments.

AR: We are very grateful to the reviewer's thorough comments.

**Specific comments:**

Line 59: Here or somewhere else in the intro, it would be good to include the explanation for how and where $CO_2$ emissions come from during the coal extraction process, as motivation for their inclusion in this study. (Currently I see that one sentence had been added to Section 3.5, following a question I raised in the first round of reviews, but would be good to see that here, as well, and expanded on a bit.)

AR: Thanks for the good suggestion. We have added the following sentence in the introduction: "Meanwhile, the extraction of coal deposits is accompanied by emissions of other non-methane gases, including $CO_2$ (Swolkien, 2020). However, $CO_2$ emissions from coal mining are usually insignificant in terms of radiative forcing when compared with $CH_4$ emissions, and are therefore rarely quantified (Bonetti et al., 2019)."

Line 97: Perhaps there should be some reference what comes out of the $CO_2$ analysis, too, if it is to be included in the manuscript?

AR: Yes, we have added two references in the previous replies. We have also added the following sentence at the end of introduction:

"Finally, we estimated both shaft-based and regional $CO_2$ emissions through the observed correlation between $CH_4$ and $CO_2$ concentrations."

Line 114: Somewhere in this paragraph, can the authors include a sentence or two about the time of day the measurements were taken and how well-mixed the atmosphere would be expected to be, including how that may affect the expected measurements?

AR: We have added the following sentence in this paragraph:

"The majority of the flights were operated between 9:00 to 14:00 (Local Standard Time, LST), when a convective boundary layer was developing or developed. Turbulent mixing was expected, which can cause complicated plume motion, e.g., meandering, a challenge for daytime measurements."

Line 170: If the difference between the on-board met station and the off-site meteorological station was this high, was that incorporated in the uncertainty calculation for the plume calculations (in both approaches) for the flights that did not have the on-board met station (flights 5-33)?

AR: Indeed, the use of off-site meteorological measurements may introduce large uncertainties; however, we do not have robust information to estimate the introduced uncertainties. Instead, we have clearly indicated in the manuscript that the uncertainty due to the use of the off-set meteorological measurements was not quantified.

Line 291: Can the authors include some examples for what causes optimization failures here?

AR: We have added the following examples after the sentence "The missing quantifications from the IG method for some flights are entirely due to failures of the optimization.":

"For example, observed concentrations on adjacent flight tracks are inconsistent due to plume meandering in one flight, as is shown in Fig. B1 #9, making it impossible to find an optimized set of parameters within their reasonable boundaries."

Line 310: It may be good to mention here that attempts were made unsuccessfully to track down the real answer here, which is the reason for making this assumption.

AR: Thanks for your suggestion. We have changed the sentence "We assume this was due to a malfunctioning $CH_4$ sensor inside the ventilation shaft" to

"We did not receive any specific explanation to the missing data, and assume this was due to a malfunctioning $CH_4$ sensor inside the ventilation shaft".

Lines 350-357: Would be a good idea to specify in the caption descriptions which are the shaft-averaged measured emissions and which are the shaft-averaged inventory emission estimates. Otherwise using just "shaft-averaged emissions" for both axes gets confusing for figures b), c), and d).

AR: We have changed to "(a) shaft-averaged quantified emissions over multiple days vs. annual coal mine emissions from the E-PRTR 2018 (Gałkowski, 2021) inventory; (b) daily shaft-averaged quantified emissions vs. daily high frequency (hourly) shaft-averaged inventory; (c) shaft-averaged quantified emissions over multiple days vs. shaft-averaged high frequency (hourly) inventory over the same days;"

Figure 8: Please specify with a legend which fit is being displayed here with the red line. Additionally, are the $R^2$ values really the same between c) and d)?

AR: We have added the description "The red lines indicate linear fits and the parameters are showed in the title" in the caption.

The same $R^2$ values for c) and d) were a mistake, and we have corrected the $R^2$ values for d).

[Figure]

Line 378: I would like to see the authors include some additional possible explanations for the systematically low estimates here. Couldn't they also result from uncertainties in the quantification approaches or some poorly-accounted-for physical process? For example, the potentially large uncertainties in the wind speed and direction (especially for flights 5-33 that did not have truly local met data)? Or perhaps non-Gaussian plume behavior due to local turbulence?

AR: First, there are also large uncertainties in the hourly inventory so that we may not be able to conclude systematically low estimates. The main reasons for the discrepancy could be the following:

1. The limited number of measurements (a total of 5 shafts and 34 available flights) and the variability of the hourly inventory (shown in Figure 7) may be the main sources of discrepancy between the quantified and inventory values. Because each flight lasts about 10 min (Section 2.1), the quantified values may not match the daily averaged hourly inventory values. Therefore, we have pointed out the potential reason "This could be due to a lack of statistics in the number of quantifications or possible biases of the measured hourly inventory."

2. As for the uncertainties for the two estimate methods, the mass balance approach is limited by the measurement time and range, and the inverse Gaussian approach may suffer from non-Gaussian plume behavior due to local turbulence and lack of temporal average, which are both quite challenging and further study is needed.

We have added the following sentence at the end of section 3.3:
"As for the uncertainties for the two estimate methods, the mass balance approach is limited by the measurement time and range, and the inverse Gaussian approach suffers from non-

Gaussian plume behavior due to local turbulence and lack of temporal average, which are quite challenging and further study is needed."

Lines 388-9: Could there be some indication either in the text or in the appendix of which of the flights were kept for this analysis? Looking at the figures, I can take a pretty good guess about which ones were probably thrown out, but it would be better to state it plainly.
AR: Thanks for the good suggestion. We have added
in the B5 caption "Flight #2/5/10/11/14/15 are used to derive $CO_2$ emissions fulfilling $R^2 > 0.5$ and the flight selection criteria.";
in B6 caption "Flight #17/18/19/21/26/30 are used to derive $CO_2$ emissions fulfilling $R^2 > 0.5$ and the flight selection criteria.";
in B7 caption "Flight #34/37/38/39/40 are used to derive $CO_2$ emissions fulfilling $R^2 > 0.5$ and the flight selection criteria.";
in B8 caption "Flight #48/49/50/51/52/53/54/58 are used to derive $CO_2$ emissions fulfilling $R^2 > 0.5$ and the flight selection criteria.".

Lines 396-397: It's still not clear to me whether this is claiming that the $CO_2$ and $CH_4$ trends themselves are similar, which I would think would be baked into the use of the linear relationship for deriving the $CO_2$ values, and thus not a surprise.
AR: Indeed, this is due to the use of the linear relationship. We have changed the sentence "Expectedly, the $CO_2$ follows the same trend as the $CH_4$, seeing strong variations on a day-to-day basis" to
"Expectedly, the $CO_2$ estimates also show strong variations on a day-to-day basis, as is for the $CH_4$ estimates."

Figure 10: Can the authors confirm that their uncertainty estimates incorporate the uncertainty associated with the CH4/CO2 slope? The 63% uncertainty on that linear fit slope is quite large, so I'm surprised to see some small error bars in the figure, e.g. in the Pniowek V chart.
AR: We have replied to this in our previous response, and we used a different method to estimate the uncertainties.

Line 456: Can the authors articulate any other potential explanations for why the values presented here appear to be lower than in Fiehn et al.? (Maybe seasonality of emissions, for example?)
AR: Our observations were performed in May and June, the same season as in Fiehn et al.,2020 and Kostinek et al., 2021. We have described the main reasons for the discrepancy in our responses above.

Line 481: Would specify for clarity "Estimated regional $CO_2$ emissions *from these coal mines*"
AR: We have added "from these coal mines" in the revised version of the manuscript.

Line 517: I believe this should read that the "coal mine ventilations shafts **are** a minor contributor to the regional $CO_2$ emissions".
AR: We have changed it.

Lines 519-526: Maybe should include reference to seasonal changes, as well, and thoughts about the representativeness of 2 weeks out of the year to upscale an entire year's estimates.
AR: Unfortunately, we do not have specific information on the impact of seasonal changes on emissions in this region, but we agree that it is necessary to point out the temporal change of the emissions.

We have added "Unfortunately, we do not have specific information on the impact of seasonal changes on emissions in this region, and we are aware that " before "short-term flights over the span of two weeks are used to estimate an annual average, where emission rates may vary week-to-week."

Figures B1-B4: Can some indicator be included to highlight which flights were included vs. excluded from the different analyses?  (Or if that's too much trouble, can this be stated in the text of the figure captions?)
AR: We have added following descriptions which flights are included in the captions of B1-B4.
B1: "Flight #1/4/13 are excluded according the flight selection criteria";
B2: "Flight #20/22/23/24/25/27/28/29 are excluded according the flight selection criteria";
B3: "Flight #32/33/41//42/43/44/45 are excluded according the flight selection criteria";
B4: "Flight #46/47/53/55/56/57/59 are excluded according the flight selection criteria".

Figures B5-B9: Similarly, can some indicator be included to highlight which flights were included vs. excluded in the analyses?  Or can this be stated in the text?  Additionally, please add the corresponding R2 values to each figure.  And why are flights 31 and 56 missing?
AR: We have illustrated which flights were included seeing answers to Lines 388-9. We have added $R^2$ values to each figure. Flights 31 and 45 are missing because of the lack of $CO_2$ information and we have added "Flights 31 is missing because of the lack of $CO_2$ information" and "Flights 56 is missing because of the lack of $CO_2$ information" in the captions of B7 and B8 respectively.

**Technical comments:**

Line 23: Please define "IG" and "MB" before using the abbreviations
We have changed to "inverse Gaussian" and "mass balance".
Line 67: Need a space between "Turnbull" and "et" in the citation
Done.
Line 104: There is a blue period and quotation mark here
Changed to black and the quotation is removed.

Line 109-110: This sentence is repetitive of line 104

We have removed it.

Line 112-113: This sentence about the "curtain" is also repeating what was just said a few lines above

We have deleted the sentence "The flight pattern for the flights…at different altitude levels" and changed "this pattern" to "this flight pattern" in the next sentence.

Line 180: "downwind of the plume"

Done.

Line 283: "presented in the next section"

Done.

Line 355: There needs to be a space between "of" and "active"

Done.

Line 378: I'm not sure what happened to the spacing between the letters in the phrase "or the possible biases of"…?

Done.

Line 387: The $M_{CO_2}$ subscript needs to be fixed

Done.

Line 393: Unpaired quotation mark at the end of the line

We have removed it.

Line 439: There is no need to include the "a" in "Fig. 11a"

Deleted.

Figure 11: The caption is counting the bars wrong after the second bar. It seems like the dark and light yellow were both counted as one bar, so all subsequent bars are described as one less than they really are (e.g. teal is actually bars 4 and 5, etc.)

We have corrected it.

Line 489: "an MB approach"

Done.

Line 504: Again, the $ppm_{CO_2}$ subscripting looks weird

Done.

Line 524: Again, the spacing between the letters of the phrase "a single goal mine" does not match the rest of the text

Done.

Line 534: "complex" not "complexed"

Done.

Line 539: "of an individual shaft"

Done.

Figures B1-B4: Please either move the x-axis label to the bottom of the last row of figures or include it in all figures

Done.